# A Lagrangian-based Floating Macroalgal Growth and Drift Model (FMGDM v1.0): application to the Yellow Sea green tide

Fucang Zhou[1], Jianzhong Ge[1,2], Dongyan Liu[1,2], Pingxing Ding[1,2], Changsheng Chen[3], Xiaodao Wei[2]

[1]State Key Laboratory of Estuarine and Coastal Research, East China Normal University, Shanghai, 200241, China

[2]Institute of Eco-Chongming, No.20 Cuiniao Road, Chenjiazhen, Shanghai 202162, China

[3]School for Marine Science and Technology, University of Massachusetts-Dartmouth, New Bedford, MA 02744, United States

*Corresponding author: Jianzhong Ge, jzge@sklec.ecnu.edu.cn

**Abstract.** Massive floating macroalgal blooms in the ocean result in many ecological consequences. Tracking their drifting pattern and predicting their biomass are essential for effective marine management. In this study, a physical–ecological model, the Floating Macroalgal Growth and Drift Model (FMGDM), was developed. Based on the tracking, replication, and extinction of Lagrangian particles, FMGDM is capable of determining the dynamic growth and drift pattern of floating macroalgal, with the position, velocity, quantity, and represented biomass of particles being updated synchronously between the tracking and the ecological modules. The particle tracking is driven by ocean flows and sea surface wind, and the ecological process is controlled by the temperature, irradiation, and nutrients. The flow and turbulence fields were provided by the unstructured grid Finite-Volume Community Ocean Model (FVCOM), and biological parameters were specified based on a culture experiment of *Ulva prolifera*, a phytoplankton species causing the largest worldwide bloom of green tide in the Yellow Sea, China. The FMGDM was applied to simulate the green tide around the Yellow Sea in 2014 and 2015. The model results, e.g., the distribution, and biomass of the green tide, were validated using the remote sensing observation data. Given the prescribed spatial initialization from remote sensing observations, the model was robust to reproduce the spatial and temporal developments of the green tide bloom and its extinction from early spring to late summer, with an accurate prediction for 7–8 days. With the support of the hydrodynamic model and biological macroalgae data, FMGDM can serve as a model tool to forecast floating macroalgal blooms in other regions.

## 1 Introduction

Floating macroalgae, primarily brown algae and some green algae, extensively occur in oceans. Except for some pelagic species, like *Sargassum*, most floating macroalgae grow in the intertidal zone during their early life stages (Rothäusler et al., 2012). Massive floating macroalgal blooms have frequently recurred in many coastal regions worldwide (Smetacek and Zingone, 2013), causing deleterious effects on economic activities and ecosystems in affected coastal areas (Lyons et al., 2014; Teichberg et al., 2010).

Some floating macroalgal bloom outbreaks seasonally, like the *Sargassum* originating from the Gulf of Mexico and the green tide in the Yellow Sea (YS), China (Gower and King, 2011; Liu et al., 2009). Under a suitable temperature and solar radiation environment, the bloom primarily begins in spring every year. Then it is advected into the adjacent sea, growing rapidly in the subsequent floating life stages until it dies. The biomass of floating green tide in the YS can exceed one million tons in late June (Liu et al., 2013; Song et al., 2015). The field and remote sensing observations are used to detect the blooming process of floating macroalgae. Field samplings, however, exhibit site-limitation and are costly, which are difficult to determine the

overall spatial development of macroalgal blooms in the regional sea (Liu et al., 2015). Remote sensing techniques can effectively estimate the coverage and quantify the total biomass (Hu et al., 2019; Wang and Hu, 2016), but they are usually challenging to capture the development and decay process owing to technical limitations and cloud cover (Keesing et al., 2011). Timely assessment and accurate prediction of coverage and biomass are essential for managing and preventing floating macroalgal bloom.

Numerical simulation is one of the most cost-effective methods to forecast spatiotemporal variations of locations and biomass for floating macroalgae. Using a numerical hydrodynamic model, we can trace the drift trajectory of floating macroalgae (Lee et al., 2011; Putman et al., 2018). The biomass, growth, and spatial coverage of the floating macroalgae change dynamically over time, which can be simulated by a biogeochemical and ecosystem model (Lovato et al., 2013; Perrot et al., 2014; Sun et al., 2020). The growth and mortality are controlled by changing environmental factors, such as temperature, light intensity, salinity, dissolved nutrients, dissolved oxygen, seawater turbidity, and predation by zooplankton (Cui et al., 2015; Shi et al., 2015; Xiao et al., 2016). Incorporating physical drifting models and the biogeochemical growth model appears essential to high-precision simulation (Brooks et al., 2018). For efficient management and forecasting, such a coupled physical and ecological model must be capable of predicting spatiotemporal variations of floating locations and biomass (Wang et al., 2018). In this study, we developed a coupled Floating Macroalgal Growth and Drift Model (FMGDM) for floating macroalgal. This model considers the influence of environmental factors, such as temperature, light intensity, and nutrients, on macroalgal growth and depletion. Driving by the flow and turbulence fields output from the Finite-Volume Community Ocean Model (FVCOM) and parameterized with sufficient physiological data, FMGDM was used to simulate the drift and growth process of the recurrent green tides in the YS during the summer of 2014 and 2015. The model was validated via satellite-derived and field-sampled data, and results were robust.

The rest of this paper is organized as follows. In Section 2, the development of FMGDM, data sources, and numerical methods are described. In Section 3, the physical and ecological driving processes are discussed, with a skill assessment of the particle tracking trajectories through the comparison with drogue-drifters and an evaluation of model reality and accuracy in the evolution process of the green tide bloom using satellite data. In Section 4, the uncertainties and prospects of FMGDM are discussed. Major innovations of this model are summarized in Section 5, following by proposed improvements of the model codes and dynamics.

## 2 Methodology

### 2.1 Model framework

The model system for FMGDM v1.0 consisted of a Lagrangian particle-tracking module and an ecological module for macroalgae growth and mortality (Fig. 1). The floating drift process is described by the Lagrangian tracking module, which is developed based on the FVCOM v4.3 offline Lagrangian tracking model (http://fvcom.smast.umassd.edu/) (Chen et al., 2012; Chen et al., 2013; Chen et al., 2021) and driven by surface wind and ocean flows. By contrast, in the ecological module of macroalgae, the processes of dynamic growth and mortality in the floating state are exhibited by particle replication and disappearance, and either the growth or mortality rate of each simulated particle is dynamically determined by the temperature, irradiation, and nutrients where the floating particle is in space and time. The position, velocity, quantity, and represented biomass of particles are synchronously updated between the two modules. The physical and environmental factors are updated from the regional and local weather, ocean numerical model system, and marine atlas datasets. Based on the updated locations and biomasses of simulated particles, the coupled individual-based tracking and ecological model, applicable in the coverage and biomass simulation of floating macroalgae, is achieved.

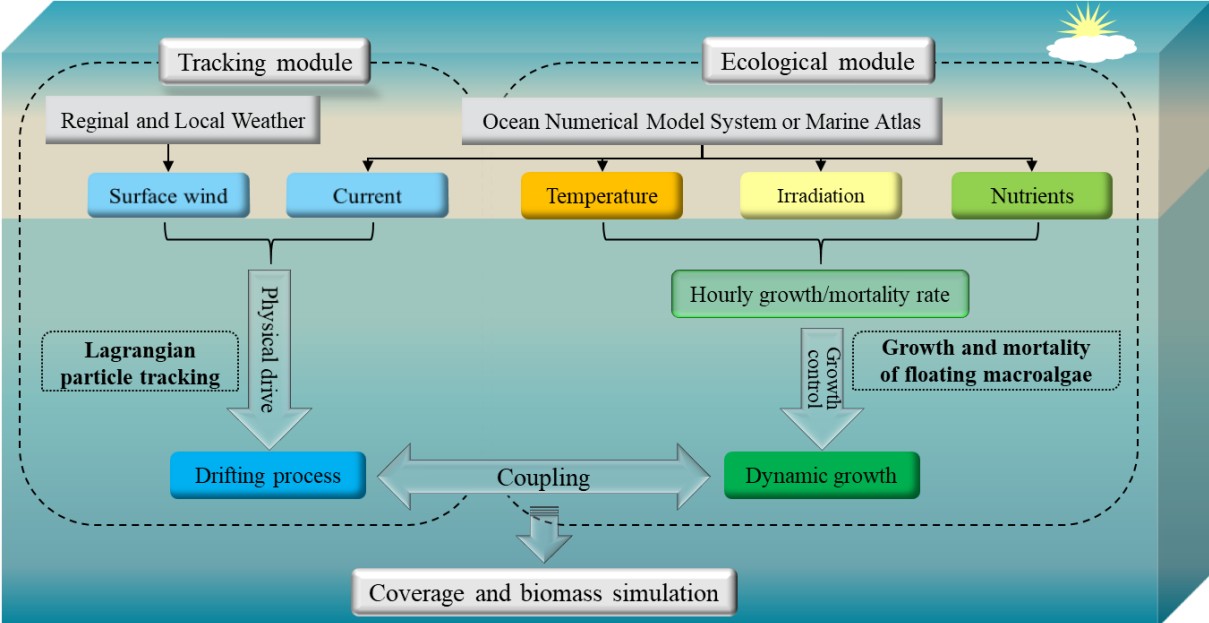

Figure 1. The framework of the physical–ecological coupled model FMGDM v1.0.

## 2.2 Lagrangian particle-tracking module

Based on the hydrodynamic model, the Lagrangian particle-tracking module was established. The current velocity $\vec{v}$ is obtained by spatially and temporally interpolating the three-dimensional (3D) velocity field from the hydrodynamic model. Horizontal and vertical interpolations were carried out via bilinear interpolation. The 10-m-height wind velocity $\overrightarrow{V_w}$ contributed to the movement of macroalgae floating at the sea surface. The windage coefficient $\kappa$ was determined by the size of macroalgae and the floating depth on the sea. $\kappa$ is assumed to be a fixed value, which does not change with the size of macroalgae in different life stages. The drifting velocity of floating macroalgae patches is determined using Eq. 1.

$$\vec{V} = \vec{v} + \overrightarrow{V_w} \cdot \kappa \tag{1}$$

To ensure the accuracy of particle trajectory, Eq. 2 is integrated by the fourth-order Runge-Kutta algorithm (Chen et al., 2021), and the time step of calculation $\Delta t$ is 60-s.

$$X_{t+\Delta t} = X_t + \int_t^{t+\Delta t} \vec{V}(x_t, t) dt \tag{2}$$

Dispersion, which was not caused by wind or currents, is also included in the trajectory tracking module. It mainly exhibited a stochastic movement, which was considered horizontal and vertical random walks by adding extra terms to particle trajectory calculation. Since the macroalgae mainly float at the sea surface without significant vertical migration, the vertical random walk was not turned on in the model setting. The horizontal random diffusion of the particles $\Delta \overrightarrow{x_r}$ is considered in simulation as Eq. 3. The coefficient of horizontal random diffusion, $K_r$ (unit: m²/s), and the time step for random diffusion was set to 6-s according to Visser's criterion. The unit vector $\vec{a}$ takes a random direction angle, and the random number $R$, fits normal distribution, takes a value between 0 and 1.0.

$$\Delta \overrightarrow{x_r}(\Delta t) = \vec{a} \cdot R \sqrt{2K_r \Delta t} \tag{3}$$

Therefore, the final position of Lagrangian particle-tracking during one-time step $\Delta t$ can be expressed as:

$$X_{t+\Delta t} = X_t + \int_t^{t+\Delta t} \vec{V}(x_t, t) dt + \Delta \overrightarrow{x_r}(t) \tag{4}$$

## 2.3 Ecological module

The ecological module reflects the growth and extinction of macroalgae by the replication and extinction of particles. One initial particle represented a patch with fixed biomass ($m_0$) of floating macroalgae and the value could be adjusted according

to needs. It was replicated and randomly released within a 2-km radius of the original location when the represented biomass of the particle exceeded $2m_0$. The biomass of the two particles returned to the initial value $m_0$. Both particles then undergo drifting and growth/extinction processes independently (Fig. 2a). Additionally, when two nearby particles had biomass below $0.5m_0$, they were combined to form one particle with $m_0$, representing the extinction process (Fig. 2b).

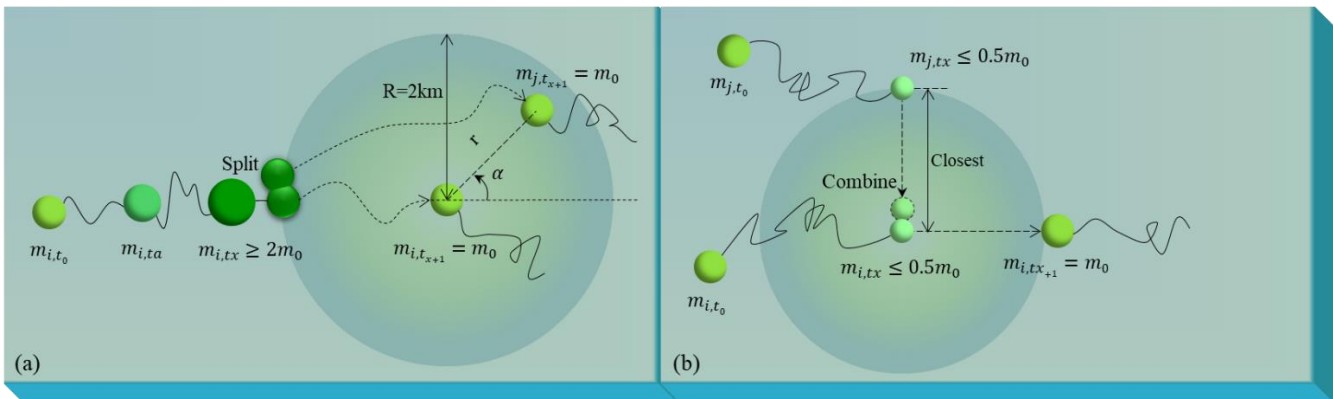

Figure 2. Diagram of the replication (a) and extinction (b) process of simulated macroalgae represented by particles.

The macroalgae growth structure of this ecological module refers to the *Ulva* sp. growth models established by Ren et al. (2014) and Sun et al. (2020). The physiological process of this module is reflected in the absorption and loss of carbon (C), nitrogen (N), and phosphorus (P). Their uptake rates were represented as $V_c$, $V_N$, and $V_P$, respectively. The loss rates were

represented as $L_c$, $L_N$, and $L_P$, respectively. The calculation of dynamic change of single-particle was expressed as:

$$\frac{dC}{dt} = V_c - L_c \tag{5}$$

$$\frac{dN}{dt} = V_N - L_N \tag{6}$$

$$\frac{dP}{dt} = V_P - L_P \tag{7}$$

We expressed the biomass evolution as carbon. The fresh weight (FW) could be determined by

$$FW = \frac{C}{K_{TOC}} \tag{8}$$

where $K_{TOC}$ indicates the conversion ratio between C and FW. Based on the physiological characteristics, the conversion value between C and FW was set as 8 mmol C /gFW (Sun et al., 2020).

The total biomass ($M_t$) of floating macroalgae throughout the domain can be determined by summing up the biomass of all

active particles.

$$M_t = \sum_{n=1}^{N_t} FW_{t,n} \tag{9}$$

The uptake and loss of *C*, *N*, and *P* are controlled by the photosynthesis, respiration, and mortality processes, proportional to biomass. The absorption of *C* is dependent on the function of photosynthesis $f(I)$, limited by the functions of temperature $f_p(T)$, nutrients $f(N)$, and light attenuation of self-shading by macroalgae $f(\rho)$.

$$V_c = f(I)f_p(T)f(N)f(\rho) \cdot C \tag{10}$$

where $f(\rho)$ is the effect of self-shading depends on the type of macroalgae and assembled density $\rho$ (mol C/m$^2$). The

photosynthesis function $f(I)$ indicates the relation between photosynthesis and irradiation $I$ (Jassby and Platt, 1976).

$$f(I) = P_{max}tanh\left(\frac{\alpha I}{P_{max}}\right) \tag{11}$$

where $P_{max}$ is the maximum photosynthetic rate and α is the photosynthetic efficiency. The changes in internal nutrients quotas have a significant impact on the physiological processes of macroalgae. The N-quota ($Q_N$) and the P-quota ($Q_P$) represented

$N{:}C$ and $P{:}C$, respectively (Ren et al., 2014). The relationship between nutrient quotas and photosynthesis is referred to Droop (1968). $Q_{Nmin}$ and $Q_{Pmin}$ are the minimum quota of N and P, respectively.

$$f(N) = min\left[\frac{Q_N - Q_{Nmin}}{Q_N}, \frac{Q_P - Q_{Pmin}}{Q_P}\right] \tag{12}$$

The nutrient uptake rate is controlled by the concentration of N and P, and limited by the functions of temperature $f_p(T)$ and absorption attenuation caused by macroalgae accumulation $f(\rho)$. The functions of nutrients uptake rate are referred to Lehman et al. (1975). The absorption of nutrients by macroalgae mainly considers dissolved inorganic nitrogen (DIN) and dissolved inorganic phosphate (DIP). The uptake rate of DIN and DIP are represented as $V_{DIN}$ and $V_{DIP}$, respectively. They are calculated as:

$$V_{DIN} = V_{mDIN}\frac{C_{DIN}}{K_{DIN} + C_{DIN}}\frac{Q_{Nmax} - Q_N}{Q_{Nmax} + Q_{Nmin}} \cdot f_P(T)f(\rho) \cdot C \tag{13}$$

$$V_{DIP} = V_{mDIP}\frac{C_{DIP}}{K_{DIP} + C_{DIP}}\frac{Q_{Pmax} - Q_P}{Q_{Pmax} + Q_{Pmin}} \cdot f_P(T)f(\rho) \cdot C \tag{14}$$

The maximum uptake rate of DIN and DIP are represented as $V_{mDIN}$ and $V_{mDIP}$, respectively. The concentration of DIN and DIP are expressed as $C_{DIN}$ and $C_{DIP}$, respectively. The half-saturation coefficient for DIN and DIP are represented as $K_{DIN}$ and $K_{DIP}$, respectively (Sun et al., 2020). $Q_{Nmax}$ and $Q_{Pmax}$ are the maximum quota of N and P, respectively.

The C loss is contributed by respiration and mortality. The C loss of respiration depends on the temperature-related function $f_r(T)$. The C loss of mortality depends on irradiance-related function $f(I)$, nutrients-related function $f(N)$ and temperature-

related function $f_m(T)$. The temperature limitation functions, $f_p(T)$, $f_r(T)$, and $f_m(T)$, corresponds to photosynthesis, respiration, and mortality processes, respectively, where $R_d$ is the dark respiration rate. When the temperature is unsuitable for the survival of macroalgae, $f_r(T)$ keeps to a minimum value indicating that minimal respiration. The mortality process replaces photosynthesis as the dominant under severe temperature and light intensity.

Similar to the loss of C, the uptake and loss of N and P are controlled by the respiration processes, and they are also proportional

to biomass. The loss rate of C, N, and P can be calculated by

$$L_c = R_d f_r(T) \cdot C + f(I)f_m(T)f(N) \cdot C \tag{15}$$

$$L_{DIN} = R_d Q_N f_r(T) \cdot C \tag{16}$$

$$L_{DIP} = R_d Q_P f_r(T) \cdot C \tag{17}$$

It should be noted that there is no interaction between this ecological module and the ocean numerical model system since this model is designed for offline computation, which is driven by the ocean model output of physical and ecosystem simulation.

## 2.4 Study area

The first green tide in the YS outbroke in 2007. Since then, it has become a recurrent phenomenon over the past 15 years

(Keesing et al., 2011; Xiao et al., 2020). The major macroalgal species involved in the green tide has been identified as *Ulva prolifera* (Ding and Luan, 2009; Duan et al., 2011). In contrast with some macroalgae that only bloom in certain areas in coastal lagoons and estuaries, the green tide, accounting for most trans-regional macroalgal blooms worldwide (Liu et al., 2013), is much more complicated, both in spatial and temporal variations. The *U. prolifera* green tide in the YS primarily originates from the coast of Jiangsu Province, primarily the coast of Yancheng and Nantong, and can drift northward to the

southern shore of Shandong Peninsula and the coastal region of the Korean Peninsula (Liu et al., 2013; Son et al., 2012) (Fig. 3). Many loosely floating propagules of *U. prolifera* were provided from mid-Apr to mid-May every year (approximately 4,000–6,000 tons), which could float and grow in the YS (Fan et al., 2015).

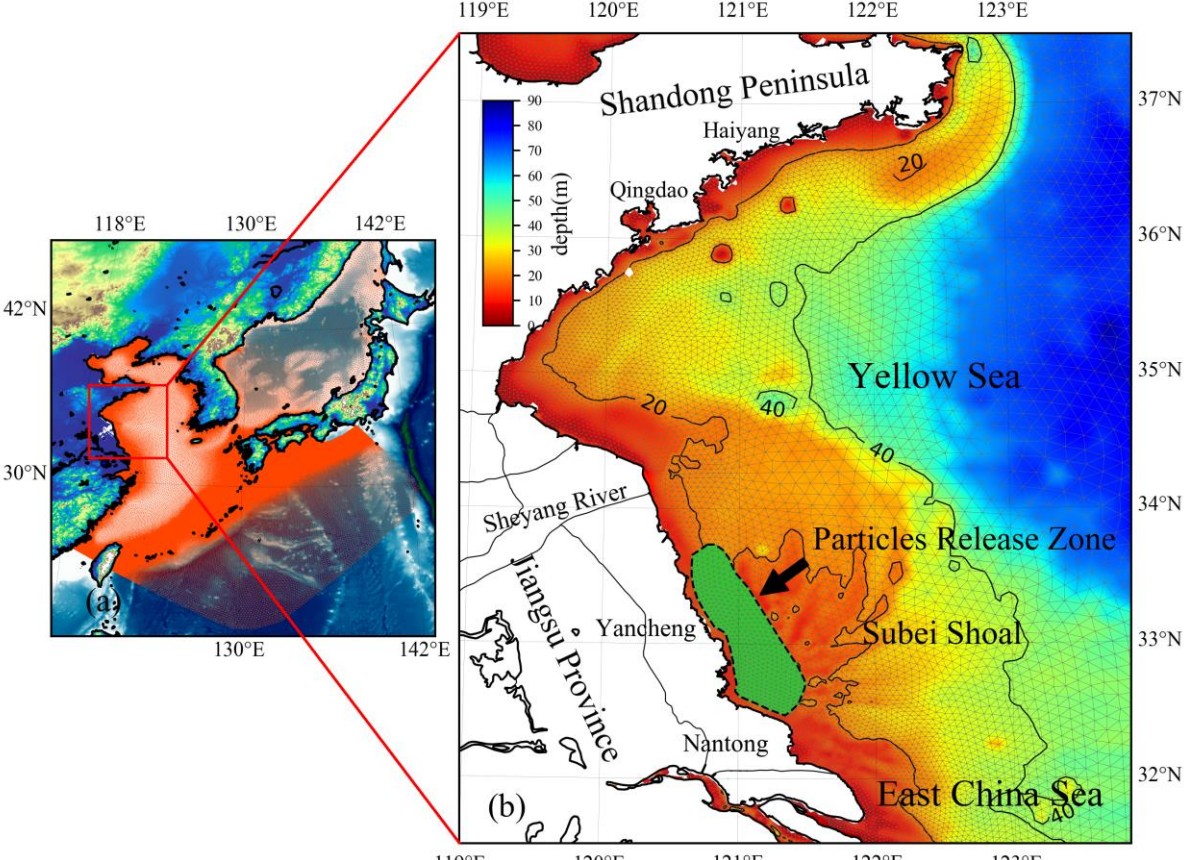

Figure 3. (a) Location of the YS, China. The red mesh indicates the high-resolution triangle grids of ECS-FVCOM. (b) The enlarged view of the area and bathymetry bounded by the solid red line rectangle in panel (a). The green area surrounded by a black dotted line in the Subei Shoal indicates the initial release zone of simulated particles.

## 2.5 Data sources

### 2.5.1 Surface wind

The wind data at 10-m above sea level derived from the surface wind dataset from the European Centre for Medium-Range Weather Forecasts (ECMWF) are available at: https://www.ecmwf.int/en/forecasts/datasets/. The wind is interpolated to the triangular grids, covering YS, East China Sea, Bohai Sea, and Japan Sea, in spatial and time scale. The interpolated wind data were used as surface forcing for ECS-FVCOM, with a spatial resolution of 0.125° and temporal resolution of 3 hours.

### 2.5.2 Satellite data

The distribution area and density of the green tide can be estimated from satellite data (Hu et al., 2019; Qi et al., 2016). In this study, the green tide's spatial distribution and growth density were validated using satellite data. The Moderate Resolution Imaging Spectroradiometer with Terra sensor (MODIS-TERRA) measured the green tide in the YS in 2014 and 2015, and the data are available from https://terra.nasa.gov/about/terra-instruments/modis. In addition, the biomass quantified based on the satellite data from Hu et al. (2019) was used to verify the simulated *U. prolifera* biomass. Blocked by clouds, remote sensing techniques exhibit difficulty in detecting small patches of floating macroalgae and often fail to capture the early status of the green tide (Garcia et al., 2013). In this study, only a few remote sensing images can be used for result verification.

The remote sensing dataset from https://www.ghrsst.org/, Group for High-Resolution Sea Surface Temperature (GHRSST), is used for data assimilation of sea surface temperature in the model system. The GHRSST dataset is daily based, with a spatial resolution of 0.01 degrees.

### 2.5.3 Drifting trajectory data

The drifting dataset used to evaluate the skills of the tracking module is composed of two parts: the trajectory data of satellite-tracked surface drifters released from the Subei Shoal in 2012 (Bao et al., 2015) and the subsurface drogue-drifters tracking data in the inner shelf of the ECS in 2017. The surface drifters contained four 40-cm width, 70-cm height rectangular sails, and a large central buoy (Bao et al., 2015). The subsurface drifter was constructed by a 67-cm diameter, 6-m height cylindrical subsurface sail, and a 28-cm diameter central buoy.

### 2.5.4 Nutrients data


The seasonal nitrate and phosphate datasets of the YS were obtained from the 1 degree-resolution World Ocean Atlas 2018 (Garcia et al., 2019) and merged with the datasets from the marine atlas of the YS (Wang et al., 1991). With combined two datasets, the nutrients at the sea surface (April to August) were applied to this simulation through temporal interpolation.

### 2.6 Hydrodynamic model

An unstructured-grid FVCOM adapts to the second-order accurate discrete flux algorithm in an integral form to solve the governing equations on an unstructured triangular grid, which provides excellent mass and momentum conservation during the calculation (Chen et al., 2006; Chen et al., 2007; Chen et al., 2003; Ge et al., 2013). To better identify the ocean circulation along with the shelf break and deep ocean, the semi-implicit discretization, which could avoid the adjustment between two-dimensional external mode and three-dimensional internal mode, was applied. With this configuration, the ocean circulation,

as well as the astronomical tide around the East China Sea (ECS), YS, and adjacent regions, could be reasonably determined (Chen et al., 2008; Ge et al., 2013). An integrated high-resolution numerical model system for the ECS (ECS-FVCOM) based on FVCOM v4.3 (http://fvcom.smast.umassd.edu/fvcom/) was established and comprehensively validated using observational data (Chen et al., 2008; Ge et al., 2013). The high-resolution triangular grid of the ECS-FVCOM domain covers the YS, ECS, Bohai Sea, and Japan Sea, which have horizontal resolutions varying from 0.5–1.5 km in the estuary and coastal region,

approximately 3 km in the path of the Kuroshio, and 10–15 km along the lateral boundary in the north Pacific region (Fig. 3a). A total of 40 layers are considered in the vertical, including five uniform layers with a thickness of 2-m specified in the sea surface and bottom to resolve better surface heating and wind mixing and bottom boundary layer (Chen et al., 2008). The ocean bathymetry was retrieved and interpolated from ETOPO1 (https://ngdc.noaa.gov/mgg/global/global.html). The initial temperature/salinity field and the volume transports along the open boundary of ECS-FVCOM were interpolated and retrieved

from HYCOM+NCODA Global 1/12° Analysis data (GLBA0.08), and eight major tide harmonic constituents (M2, S2, K2, N2, K1, O1, P1, and Q1), which are obtained from TPXO 7.2 Global Tidal Solution (Egbert and Erofeeva, 2002), were used along the open boundary (Ge et al., 2013). The freshwater discharge of the Yangtze River and Qiantang River (source: http://www.cjh.com.cn/) was added to the upstream river boundary. Surface wind and radiations from ECMWF were used in ECS-FVCOM as surface forces. In addition, the GHRSST dataset was applied to better determine the sea surface temperature

using the model-data assimilation. The simulation time was set from March 29 to September 1, thus covering early spring to late summer. The water velocity, temperature, salinity from ECS-FVCOM were fed into the FMGDM as input variables.

### 2.7 Model settings

Seven particle-tracking simulations were conducted. One hundred particles were released at a location that matched the drifter's *in situ* deployment position, and the horizontal random diffusion coefficient ($K_r$) was set as 50 m$^2$/s. In addition, the

depth for surface and subsurface drifters were set as 0.5-m and 2-m, respectively. Thus, only half of the buoy was exposed above the sea surface for these drogue-drifters, and the direct wind factor was not considered in the tacking simulations.

For the wind-exposed drifters floated at the sea surface, the wind drift is one of dominating contributions of the transportation. Dagestad and Röhrs (2019) conducted drifting buoy experiments and found that windage accounted for 3% of Stokes drift.

Whiting et al. (2020) chose a constant 3% coefficient in the free-floating macroalgal trajectory simulation. The setting of this coefficient was based on the debris drift simulation of the 2011 Japan tsunami (Maximenko et al., 2018). Additionally, Jones et al. (2016) reported that the horizontal movement of surface oil slicks is drifted by ~3.5% wind speed, including a 2% direct wind drag and a 1.5% wind speed adding to the surface Stokes drift (Abascal et al., 2009). The movements of free-floating macroalgae are influenced by wind and windage, which depend on the physical characteristics of drifters. The hydrodynamic

surface layer had already accounted for wind movement, and the other wind drag for particle drift was composed by direct windage.

Based on the previous studies described above, the *U. prolifera*-induced total drifting windage was in a range of 2.7–3.5%. A series of particle tracking experiments were conducted in this windage range, with an interval of 0.1%. Meanwhile, one experiment without the direct wind factor was also undertaken for reference. Totally ten groups, with 1,192 particles in every

group, separated with a 0.02° horizontal resolution, were deployed in batches in the particle release zone (Fig. 3b) on May 1, 2014. These particles were traced for 120 days.

Most importantly, two realistic dynamic growth simulations were conducted. To verify the general applicability of the model, we simulated the growth and drift processes of *U. prolifera* in the YS in 2014 and 2015, respectively, with identical model configurations. In the two simulations, each particle represented 10-ton biomass of floating *U. prolifera*, so that 4,800 tons

were deployed initially. The initial coverage and biomass of the *U. prolifera* were determined based on the field surveys by Liu et al. (2013) and Xu et al. (2014a). The simulation time was 135 days from April 16 to August 29. The initial particles deployed continuously from April 16 to May 15. Daily 160-ton biomass was spatial randomly released in the hot-spot zone (Fig. 3b) over an entire month. The horizontal random diffusion $K_r$ was set as 200 m$^2$/s in the green tide simulation. In this study, instantaneous environmental factors, including temperature, nutrients, solar radiation intensity, ocean flow, and wind

speed, were determined from the physical ECS-FVCOM model.

We set the parameters of the ecological module according to the physiological characteristics of *U. prolifera*. The functions of temperature were determined referring to the results of laboratory studies. *U. prolifera* has the optimal photosynthesis efficiency at 20 °C, and turns white and declines rapidly under high temperature and high light intensity (Cui et al., 2015; Song et al., 2015). When the temperature is suitable (5–25.7 °C), the temperature limitation of photosynthesis $f_p(T)$ and respiration

$f_r(T)$ are consistent. When the temperature becomes unsuitable (<5 °C or >25.7 °C), the respiration of macroalgae, unlike photosynthesis, will remain at a lower level. When the high temperature exceeds a suitable situation (>25.7 °C), the mortality process replaced photosynthesis as dominance.

$$f_p(T) = \begin{cases} -4.942 \times 10^{-4}T^3 + 0.01885T^2 - 0.135T + 0.1014, & 5°C \leq T \leq 25.7°C \\ 0, & T < 5°C \text{ or } 25.7°C < T \end{cases} \tag{18}$$

$$f_r(T) = \begin{cases} -4.942 \times 10^{-4}T^3 + 0.01885T^2 - 0.135T + 0.1014, & 5°C \leq T \leq 25.7°C \\ 0.789, & T < 5°C \text{ or } 25.7°C < T \end{cases} \tag{19}$$

$$f_m(T) = \begin{cases} 0, & T \leq 25.7°C \\ 0.01416T^3 - 1.223T^2 + 35.22T - 337.73, & T \geq 25.7°C \end{cases} \tag{20}$$

According to the floating growth characteristics of *U. prolifera*, the self-shading limited function $f(\rho)$ was determined. When the assembled density does not exceed 0.16 mol C/m$^2$, the growth of *U. prolifera* is not restricted by self-shading. However,

as the density increases, the accumulation of *U. prolifera* becomes significant, and maximum when the density is greater than 0.56 mol C/m$^2$.

$$f(\rho) = \begin{cases} 1, & \rho \leq 0.16 \\ 2.308 \exp(-2.5\rho) - 0.54705, & 0.16 < \rho \leq 0.56 \\ 0, & \rho > 0.56 \end{cases} \tag{21}$$

The complete list of parameters used in the ecological module of *U. prolifera* was shown in Table. 1.

Table. 1. Parameters used in ecological module for *U. prolifera*, modified from Sun et al. (2020).

| Parameters | Description | Value | Dimension | Reference |
|---|---|---|---|---|
| $Q_{Nmin}$ | Minimum N-quota | 25.3 | $mmol\ N\ mol\ C^{-1}$ | (Fujita, 1985) |
| $Q_{Nmax}$ | Maximum N-quota | 108.7 | $mmol\ N\ mol\ C^{-1}$ | (Sun et al., 2015) |
| $Q_{Pmin}$ | Minimum P-quota | 0.097 | $mmol\ N\ mol\ C^{-1}$ | (Sfriso et al., 1990) |
| $Q_{Pmax}$ | Maximum P-quota | 1.4 | $mmol\ N\ mol\ C^{-1}$ | (Sun et al., 2015) |
| $V_{mDIN}$ | Maximum uptake rate of DIN | 2.8 | $mmol\ N\ mol\ C^{-1}h^{-1}$ | (Li and Zhao, 2011; Luo et al., 2012b) |
| $V_{mDIp}$ | Maximum uptake rate of DIP | 0.58 | $mmol\ N\ mol\ C^{-1}h^{-1}$ | (Luo et al., 2012b) |
| $K_{DIN}$ | Half-saturation coefficient for DIN | 18.77 | $\mu mol\ N\ L^{-1}$ | (Li and Zhao, 2011; Luo et al., 2012b) |
| $K_{DIP}$ | Half-saturation coefficient for DIP | 10 | $\mu mol\ N\ L^{-1}$ | (Luo et al., 2012b) |
| $P_{max}$ | Maximum photosynthetic rate | 240.51 | $\mu mol\ C\ g\ WW^{-1}h^{-1}$ | (Xu et al., 2014b) |
| $\alpha$ | Photosynthetic efficiency | 2.52 | $(\mu mol\ C\ g\ WW^{-1}h^{-1})$ $/(\mu mol\ photons\ m^{-2}s^{-1})$ | (Xu et al., 2014b) |
| $R_d$ | Dark respiration rate | 18.4 | $\mu mol\ C\ g\ WW^{-1}h^{-1}$ | (Xu et al., 2014b) |

## 3 Results

### 3.1 Variations of environmental factors

#### 3.1.1 Surface wind

The wind vectors at 10-m-height near Subei coast and Qingdao coast, retrieved from ECMWF, were showed in Figure 4. From May to July 2014, southerly and southeasterly winds prevailed in the coast of Subei and Qingdao, and the mean wind speed reached 5 m/s. However, the southerly wind was stronger throughout May in Spring. In June, southeast winds blew in the Subei coast significantly, and the southerly wind still dominated the Qingdao coast; the wind speed was slightly lower than that in May. In August, the northeast wind was strengthened, especially from August 1–10.

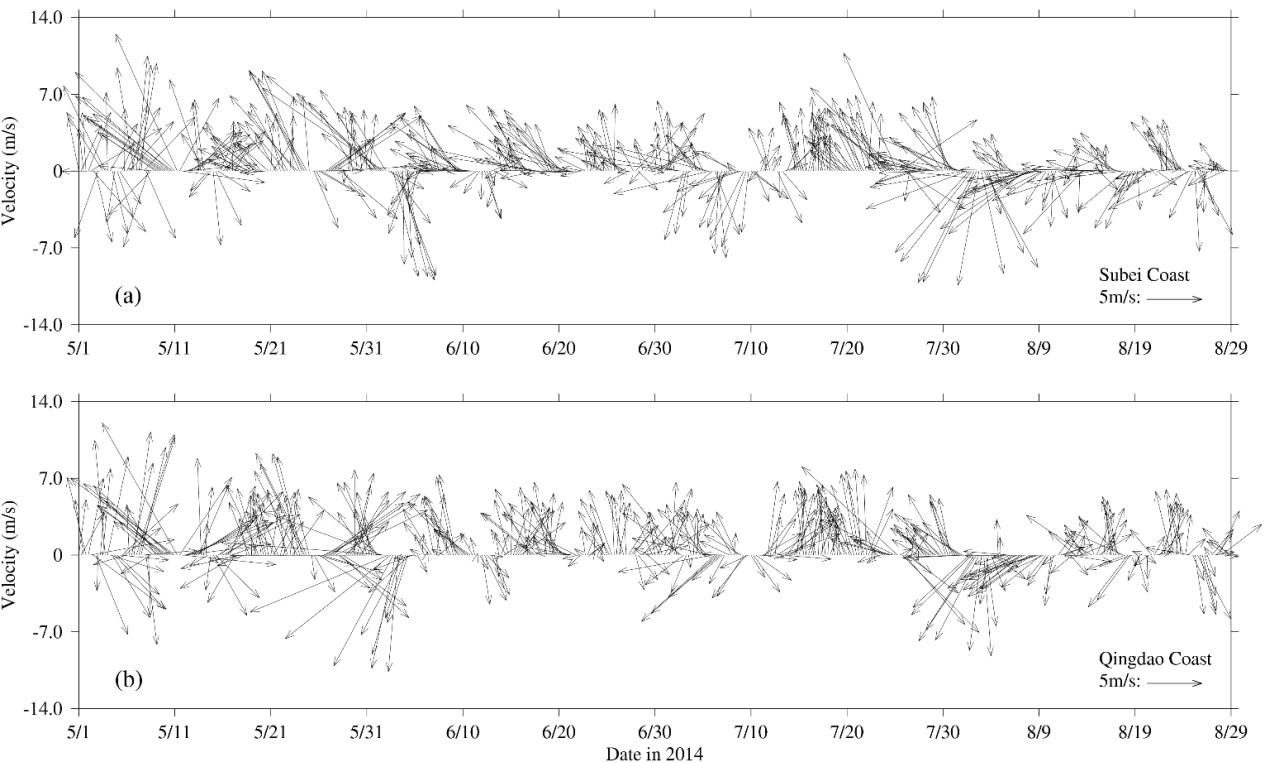

Figure. 4. Wind vectors at 10-m-height near Subei coast (a) and Qingdao coast (b) at a time interval of 6-hrs from May to August 2014.

### 3.1.2 Ocean circulation

The time-averaged distributions of surface ocean circulation every 15-d period in 2014 and 2015 are shown in Figs. 5 and 6, respectively. Affected by the southerly and southeasterly winds (Fig. 4), the coastal surface seawater flowed northward and was transported to the east of South YS. This phenomenon was more pronounced in May and late June, and July (Fig. 5a, b, d, f), indicating the possibility of *U. prolifera* drifting from the Subei Shoal toward the north. The same phenomenon was observed from May–June and early August of 2015 (Fig. 6a–d, g). Most of the time, surface seawater from north YS was transported to South YS through the east of Rongcheng (RC). In early June and July, and August 2014 (Fig. 5c, e, g, h), the surface seawater circulated counterclockwise in the middle region of South YS. Simultaneously, the weak currents on the south side of the Shandong Peninsula may have caused *U. prolifera* to stay in this region and gradually land on the shore. Similar ocean circulations appeared in July and late August of 2015 (Fig. 6e, f, h), and weak northward currents were observed in the Subei Shoal.

### 3.1.3 Temperature

Figures 5 and 6 also included the distribution of surface seawater temperature in 2014 and 2015, respectively, with every 15-d time averaging. The average temperature of South YS reached 13 °C in May (Fig. 5a, b) and increased continuously in June (Fig. 5c, d). Surface seawater temperature along Jiangsu Coast and the East China Sea was generally 1–2 °C higher than that in other areas of South YS. However, most South YS reached a high temperature, with over 25 °C, by July 2014 (Fig. 5e, f). From mid-July to end-August, the surface temperature in Jiangsu Coast and parts of Shandong Peninsula Coast remained above 27 °C (Fig. 5f–h). The offshore sites of Qingdao and Subei were selected to determine the time series process for the physical factors (Fig. 5i–j and Fig. 6i–j). The surface temperatures of the two stations, the northern Jiangsu Coast and Qingdao coast, were increased until they reached their peaks at the end of July with over 27 °C and remained until the end of August. The distribution and tendency of South YS seawater temperature in 2015 (Fig. 6) were similar to those in 2014. However, compared with those in 2014, they had more extensive high-temperature coverage for South YS in August 2015 (Fig. 6g–h). The surface temperature of most YS regions exceeded 27 °C, part of the Jiangsu Coast even reached 29 °C. In addition, the surface temperatures of the two stations reached 25 °C in 2015, approximately one week later than they did in 2014 (Fig. 6i–j).

### 3.1.4 Irradiation and salinity

Solar irradiation intensity is significantly different in the day and night. Therefore, only the irradiation intensity at noon was analyzed in Fig. 5i–j and Fig. 6i–j. Affected by the thickness of clouds, irradiation intensity at noon fluctuated drastically within 3, 200 $\mu mol \cdot m^{-1} \cdot s^{-1}$. Compared with May and June, the irradiation intensity in July and August 2014 and 2015 decreased slightly.

The surface salinity of South YS fluctuated between 29 PSU and 33 PSU during the period of the green tide bloom (Fig. 5i–j and Fig. 6i–j), which was suitable for *U. prolifera* growth (Xiao et al., 2016). For this reason, the salinity limitation was ignored in the biological module.

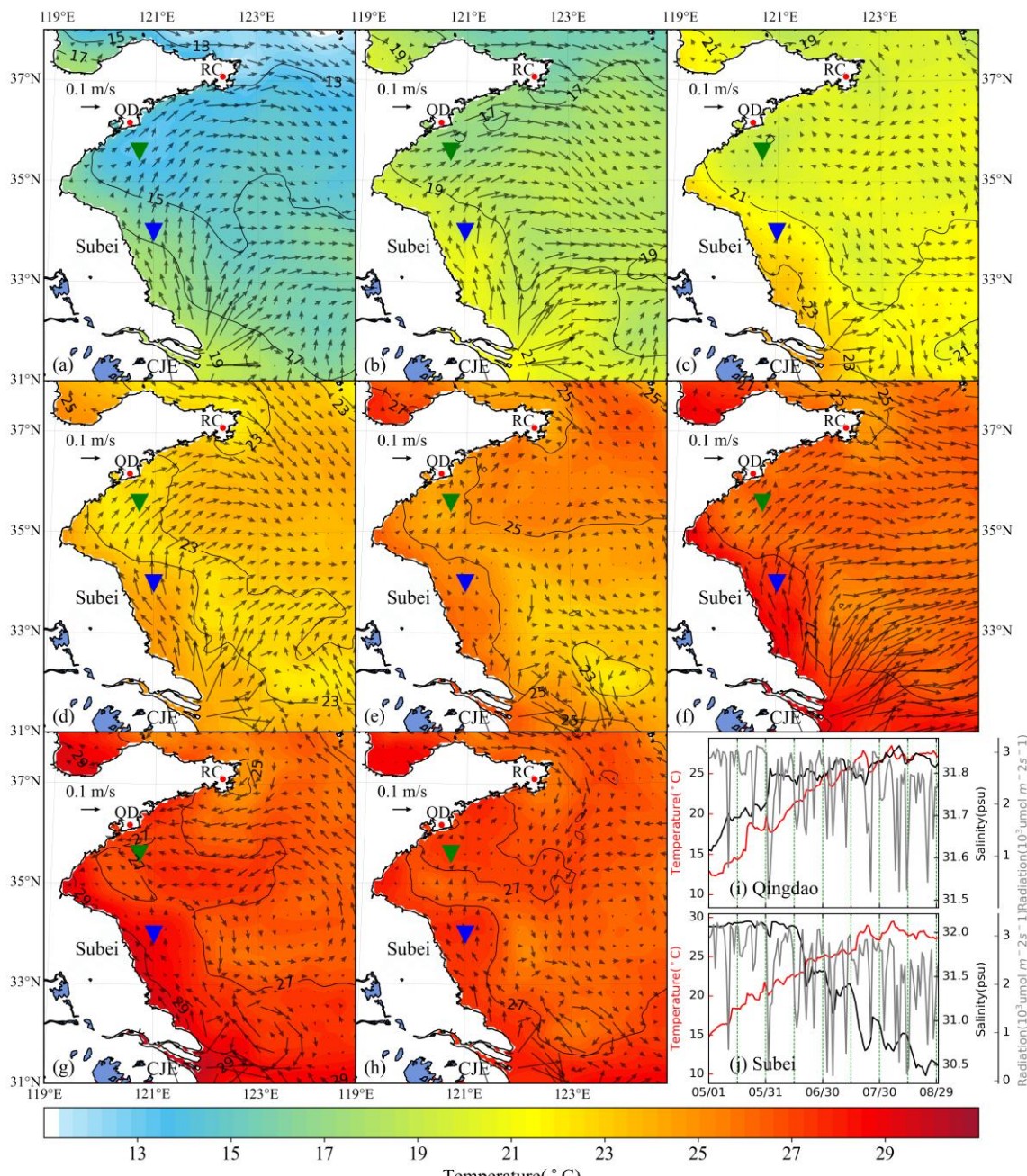

Figure. 5. Time-averaged distribution of surface current and temperature of every 15-d duration in 2014: (a) 1–15 May, (b) 16–30 May, (c) 31 May–14 June, (d) 15–29 June, (e) 30 June–14 July, (f) 15–29 July, (g) 30 July–13 August, (h) 14–29 August. The green and blue inverted triangles indicate the position of selected Qingdao (QD) coast and Subei offshore sites, respectively. Time series of surface temperature, salinity, and irradiation in Qingdao (i) and Subei (j).

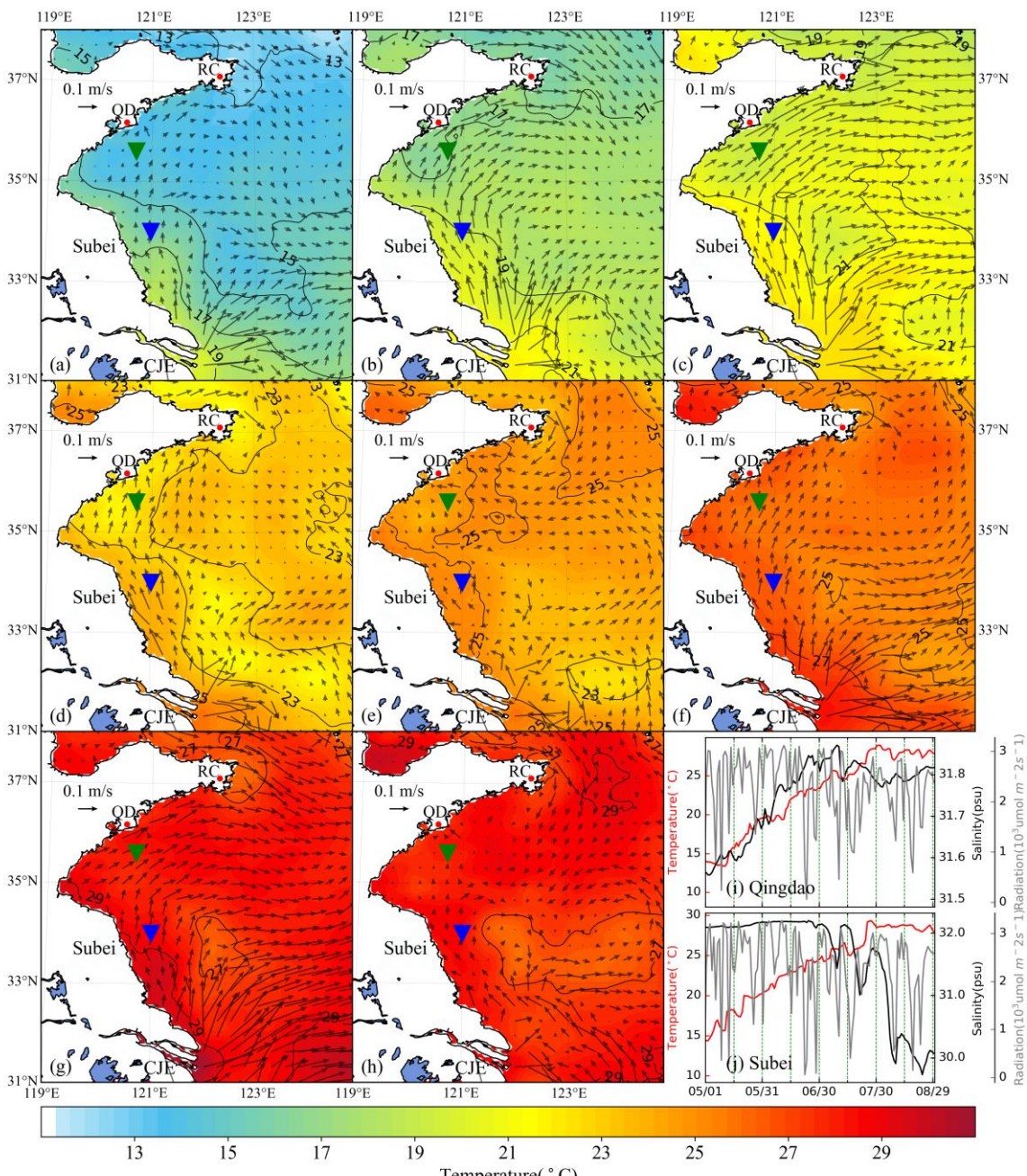

Figure. 6. Time-averaged distribution of surface current and temperature of every 15-d duration in 2015: (a) 1–15 May, (b) 16–30 May, (c) 31 May–14 June, (d) 15–29 June, (e) 30 June–14 July, (f) 15–29 July, (g) 30 July–13 August, (h) 14–29 August. The green and blue inverted triangles indicate the position of selected Qingdao (QD) coast and Subei offshore sites, respectively. Time series of surface temperature, salinity, and irradiation in Qingdao (i) and Subei (j).

### 3.1.5 Dissolved nutrients

The dissolved inorganic nutrients in the offshore region are mainly influenced by terrestrial sources, with prominent seasonal characteristics. The concentration of dissolved inorganic nutrients in the Jiangsu region was significantly higher than in other areas. The nitrate concentration in the offshore region of Jiangsu was generally above 2 mmol/m$^3$ in spring (Fig. 7a) and summer (Fig. 7b), especially in the Yancheng region; the nitrate concentration was over 8 mmol/m$^3$ in spring. The nitrate concentration in the other areas of YS, except the offshore region of Jiangsu, was mainly below 2 mmol/m$^3$. The phosphate concentration in the offshore region of Nantong and Sheyang River Estuary was still high, more than 0.6 mmol/m$^3$ in spring (Fig. 7c) and summer (Fig. 7d). In the north of the Yancheng offshore region, phosphate concentration decreased to ~0.2

mmol/m$^3$ in summer (Fig. 7d). In the central YS and south offshore area of Shandong Peninsula, phosphate concentration was higher in summer, over 0.2 mmol/m$^3$, than ~0.1 mmol/m$^3$ in spring.

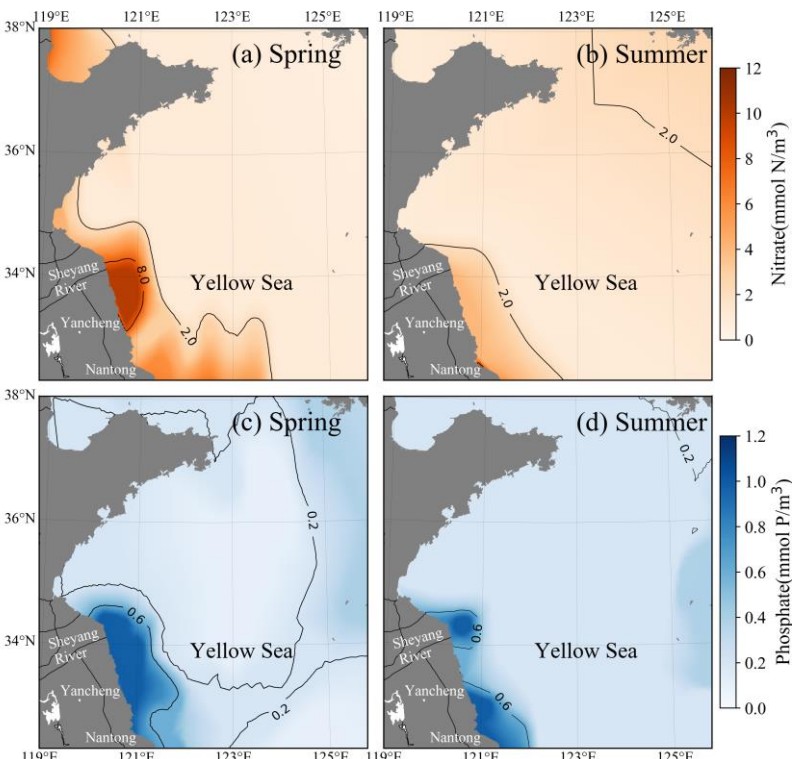

Figure. 7. Seasonal-averaged surface distributions of nitrate (upper) and phosphate (lower) in YS during spring (left), summer (right).

## 3.2 Validation of Tracking module

### 3.2.1 Tracking module evolution

The simulated particle trajectories are generally consistent with the observed drifter trajectories, particularly for a short-term prediction (Fig. 8). The tracking time for surface drifters #1–5 lasted more than one month (Fig. 8a–e). The results show that the model was robust to reproduce the overall drifter's movement directions. Since the drifter paths may change with strong randomness due to complex variations of ocean flow, winds, and waves, the long-term prediction of drifter paths and dispersal could be of great challenge. The tracking time for subsurface drifters #6–7 (Fig. 8f) was relatively shorter, only 5–9 days. Compared with surface drifters, subsurface drifters were driven by a more complicated forcing relating to the extensive depth range of the sail. The water flow at a 2-m depth was selected as the driving force approximately, considering the average drifting state of subsurface drifters. The simulated trajectories were similar to the observed drifter's movement trends. However, the drifted distance was slightly shorter than the actual situation.

The model-data compression suggests that the particle tracking algorithm in FMGDM can provide reasonable predictions for free-floating drifters with higher confidence for surface drifters. In addition, the hydrodynamic model, ECS-FVCOM, is reliable. Our results showed that *U. prolifera* were mainly under a free-floating state at the sea surface.

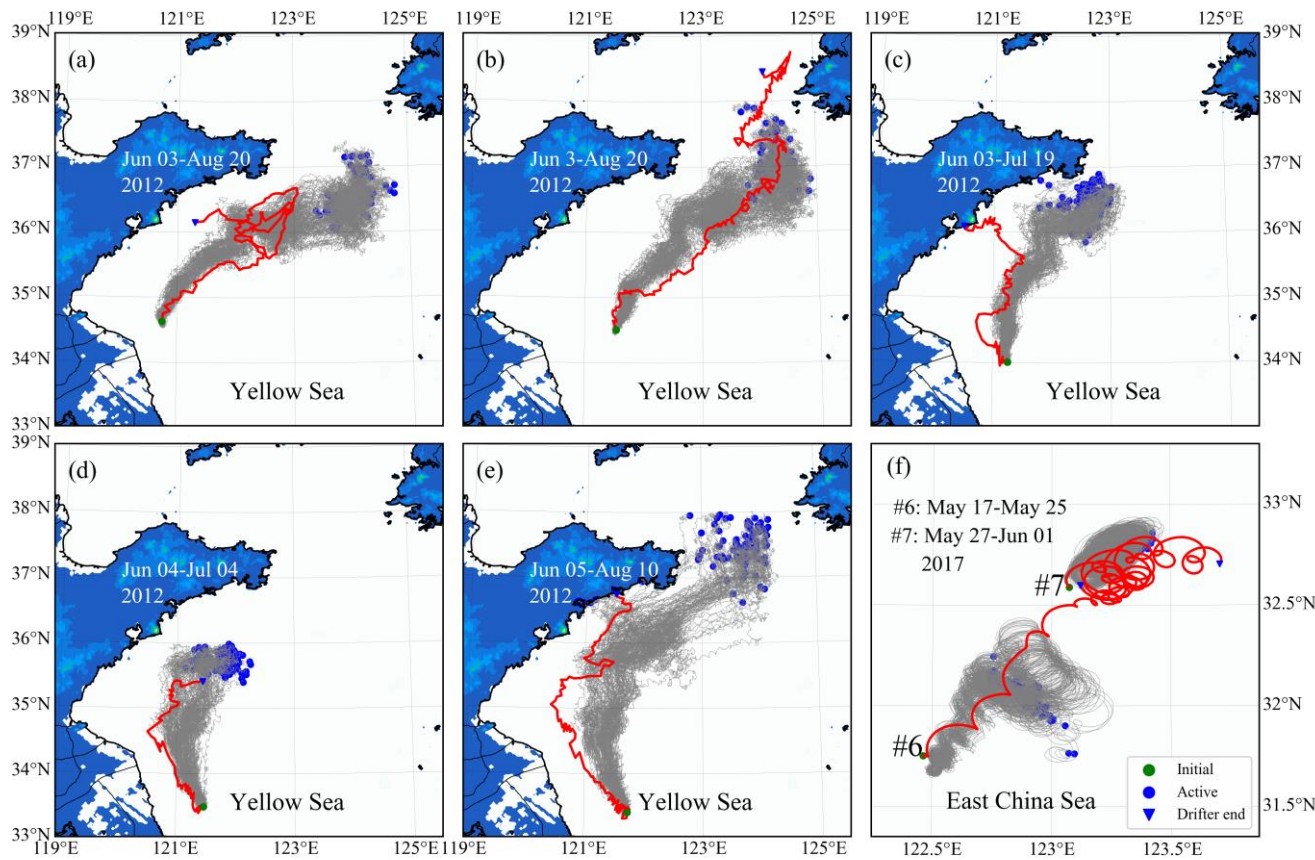

Figure. 8. Comparisons between observed (red lines) and simulated (gray lines) drifter trajectories. Panels (a–e) correspond to the drifters #1–5, respectively. Panel (f) corresponds to drifters #6–7.

### 3.2.2 Windage simulation

335   The simulation results with different windages showed that particles first flowed northward and then turned north-eastward (Fig. 9). With greater windage, the trend of northward transportation is more prominent. The particles with windage of about 3.4% could reach the southern coast of the Shandong Peninsula on June 15 (Fig. 9b) and then turned north-eastward near 124º30.00'E (Fig. 9c–d). The particle group was split at the end of July. One part drifted northward continuously to 38ºN and reached the North Korean coast, and the other was turned west (Fig. 9f). The particles with a windage coefficient less than

340   3.2% were stranded near the southern coast of the Shandong Peninsula in July and August (Fig. 9d–f). The particles without direct windage have significantly slow drifting, northernmost nearly to the south coast of Shandong Peninsula. Some particles moved north-eastward to the central of the South YS. From the comparison, winds contributed significantly to the transport of free-floating drifters. The transport results with a 2.7–3.5% windage range did not show a significant difference in the short-term simulation of 1–1.5 months (Fig. 9a–c). However, as the simulation time lasted longer, the transport pattern showed a

345   noticeable difference (Fig. 9d–f). Compared with the evolution of green tides in the YS from remote sensing (Hu et al., 2019), it can be confirmed that the windage in a range of 3–3.2% could be applied to the drift of green tide. In this study, 3.2% was selected as the windage $\kappa$ of the YS green tide simulation.

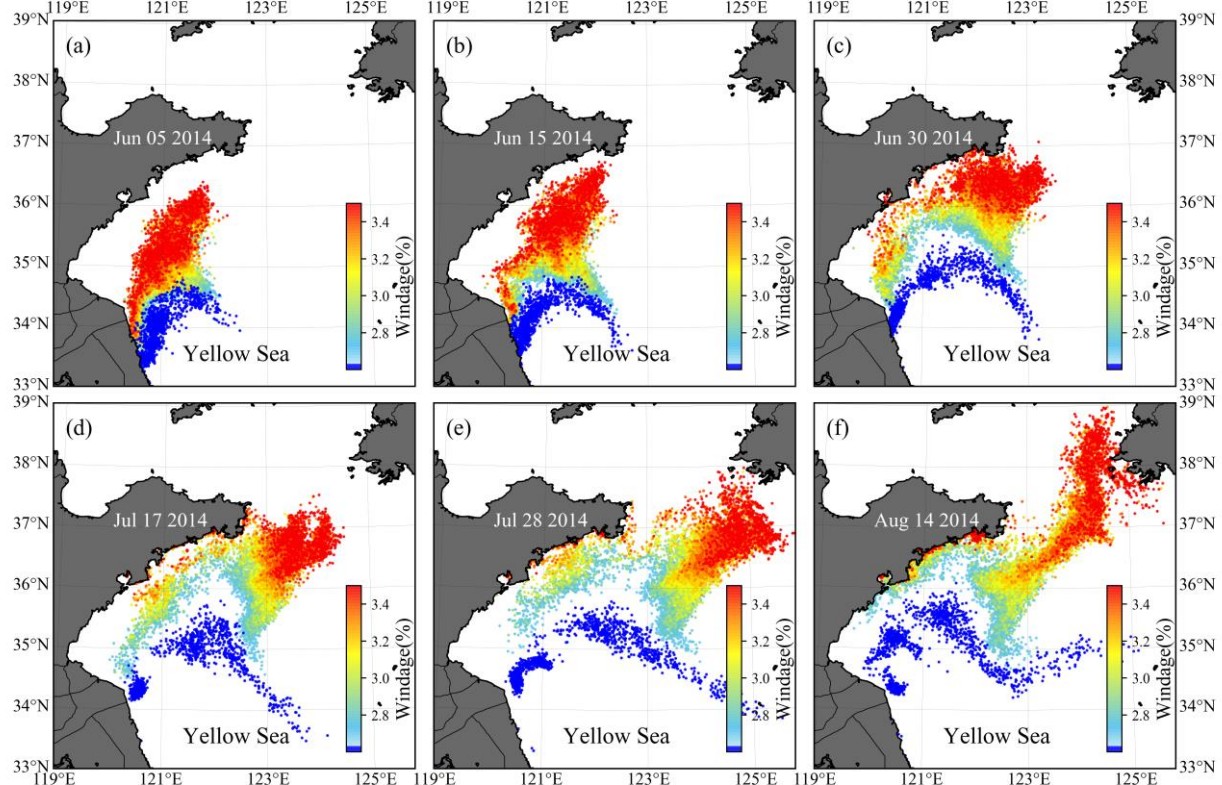

Figure. 9. Evolution of the particle distribution in FMGDM tracking simulations. Colors indicate different windage (2.7%-3.5%) of the simulated particles. The blue particles indicate the simulation without direct windage. (a) June 5, 2014; (b) June 15, 2014; (c) June 30, 2014; (d) July 17, 2014; (e) July 28, 2014; (f) August 14, 2014.

### 3.3 Simulation of dynamic growth model

### 3.3.1 Spatiotemporal variation of *U. prolifera*

After being released into the Subei Shoal, the initial particles drifted and dispersed by ocean flows and wind. The simulation result of the green tide in 2014 is shown in Fig. 10. It showed a small amount of *U. prolifera* floating on the Subei coast in mid-May (Fig. 10a). However, it was difficult to be observed using remote sensing technology in the early stage of green tide bloom. After one month of simulation, the modeling biomass increased to approximately 0.2 million tons (Fig. 10i). Both observation and simulation showed that *U. prolifera* was transported northward and floated between northern Jiangsu offshore and Shandong Peninsula (Fig. 10b). On June 15 (Fig. 10c), both the results of observation and simulation show that green tides had landed on the southern coast of Shandong Peninsula, including Rizhao (RZ) and Qingdao shore. Moreover, the observations show that the green tide bloomed around the coast of Nantong (NT) and Yancheng (YC), suggesting the continuous supply of additional *U. prolifera* from aquaculture raft between May and June 2014. On June 23 (Fig. 10d), the result of both observation and simulation were consistent and showed that green tides had landed on the Shandong Peninsula on a large scale, and the farthest *U. prolifera* reached the Rushan (RS) coast. The entire coast and offshore regions were covered with a massive amount of floating *U. prolifera.* Due to the high concentration of nutrients, there were a large number of simulated particles growing at the Sheyang River estuary region in the entire June. The biomass of simulation reached a peak of 0.85 million tons on June 30, and the number of simulation particles reached approximately 71,000 (Fig. 10i). Subsequently, *U. prolifera* died out quickly, and its coverage decreased significantly. On July 17 (Fig. 10f), the floating *U. prolifera* still gathered on the south coast of the Shandong Peninsula. Different from the observation result, some small patches of simulation result drifted eastward and reached 123°E.

In contrast with the simulation results, observation showed the re-occurrence of a large-scale green tide in Yancheng and Nantong regions from July 17 to July 28 (Fig. 10f–g), which, however, was uncaptured by the model. Both observation and simulation results showed that floating *U. prolifera* drifted eastward but still covered the south coast of the Shandong Peninsula at the end of July (Fig. 10g). After half a month, floating *U. prolifera* had died out (Fig. 10h). Observation shows that only the southern coast of Qingdao and the Subei Shoal had a few patches of *U. prolifera* on August 14, which suggests there is still a possible *U. prolifera* source near the Subei Shoal even in the summertime of July and August. However, the *U. prolifera* of simulation had almost vanished because the *U. prolifera* source was only initialized from mid-April to mid-May.

As there was no direct way of quantifying the floating *U. prolifera* biomass of green tides throughout the YS (Wang et al., 2018), the estimated biomass data of *U. prolifera* retrieved from remote sensing observations (Hu et al., 2019) was adopted to validate the simulated biomass (Fig. 10i). The estimated biomass of *U. prolifera* rose rapidly and peaked with maximum values of 0.92 million tons on June 18, 2014 (Hu et al., 2019). The biomass declined rapidly after reaching its peak, and *U. prolifera* almost died off at the end of July. Compared with observation results, the biomass of simulation peaked after 12 days with a similar value. The growth trends between observation and simulation were similar. Considering the highly random dispersion, as well as the dynamic life history of *U. prolifera*, our simulation provides reasonable modeling results of biomass and spatial coverage.

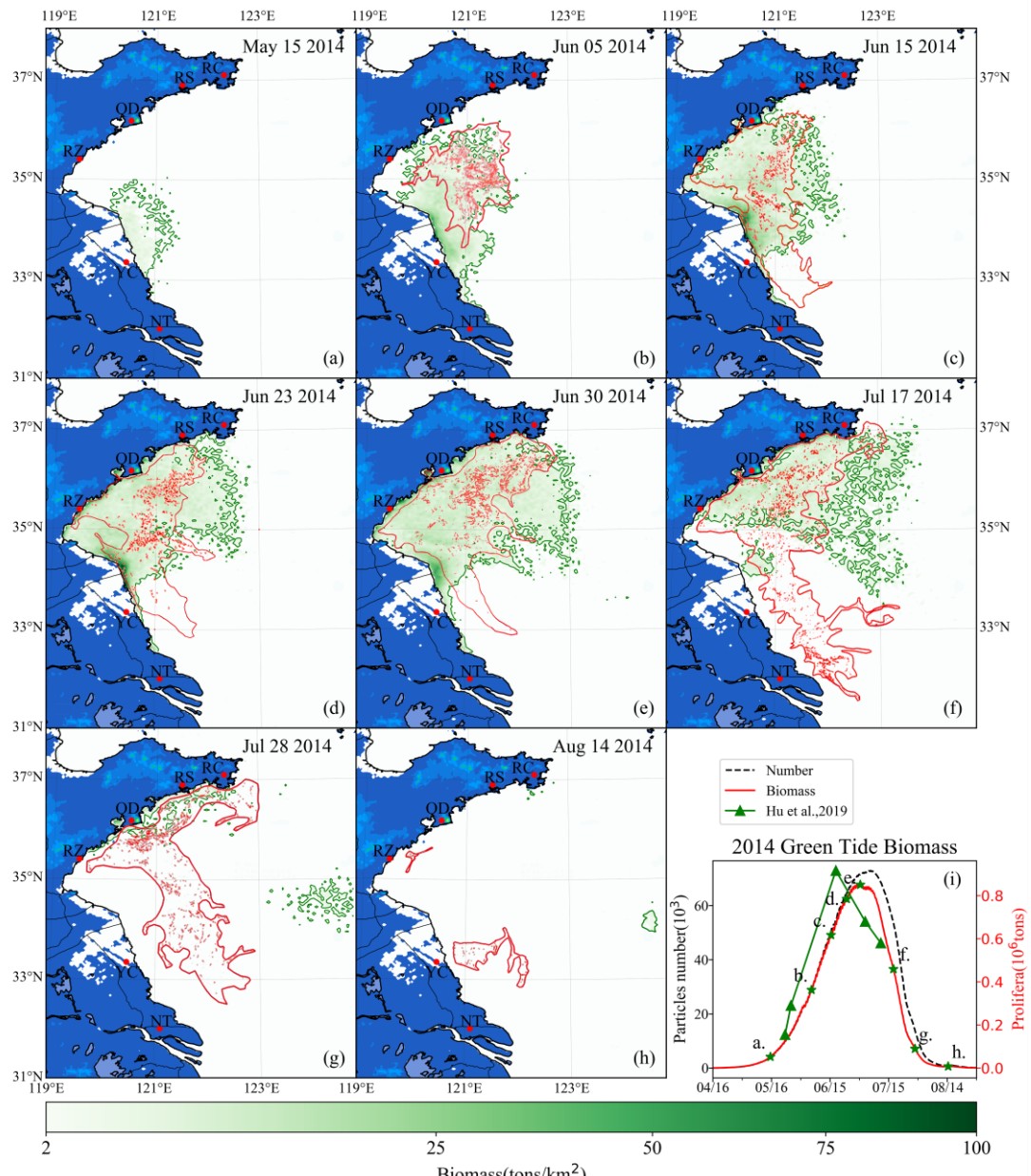

Figure 10. Comparison between simulation and remote sensing observation of green tides from May to August 2014 (a–h). The green image indicates the simulated biomass density of the green tide (color image, unit: tons/km$^2$). The red image shows the satellite-derived spatial coverage of *U. prolifera* from MODIS-TERRA. Panel (i) is the time series of simulated biomass and particle number of green tides in 2014, compared with observed biomass from Hu et al. (2019). The green pentagrams in panel (i) indicate the biomass of the corresponding date in the panel (a–h).

To verify the reliability of the coupled model system, the green tide that bloomed in 2015 was also simulated and compared with the observations made. The simulation shows that a small amount of *U. prolifera* was floated near the coast of Jiangsu in mid-May (Fig. 11a). On May 30, the coverage of floating *U. prolifera* increased, while the northernmost green tide patches were closed to 35°N (Fig. 11b). On June 23 (Fig. 11c), both the results of observation and simulation showed that the green tide had entered the Shandong Peninsula with large-scale coverage, distributed in most of the seas from Subei to the Shandong Peninsula and bloomed strongly offshore Qingdao to RS. On July 2, both observation and simulation showed that the green tide still gathered along the south coast of the Shandong Peninsula, and the northernmost of the distribution range reached RC (Fig. 11d). In addition, observation showed scattered patches of *U. prolifera* floating in the center of the South YS from June 23 to July 2, which cannot be simulated. On July 16 (Fig. 11e), satellite observations showed that the coverage of green tide

reduced considerably, and the distribution range was shrunk toward the west of 121°30'E. In late July, the simulation result showed that the biomass declined rapidly; however, the green tides are still widely distributed in the southern regions of the Shandong Peninsula. On August 5, observations showed small patches of the floating green tide in the middle of South YS (Fig. 11g). In simulation results, a small amount of the green tide remained along the coast of the Shandong Peninsula. On August 20 (Fig. 11h), the green tide in the YS completely disappeared from satellite observation and numerical simulation. Compared the biomass of observation and simulation, the estimated biomass of *U. prolifera* based on remote sensing peaked with maximum values of approximate 1.77 million tons on June 21, 2014 (Hu et al., 2019), while the simulated biomass peaked after 13 days with a similar value of 1.6 million. The number of simulation particles peaked at approximately 134,000 (Fig. 11i).

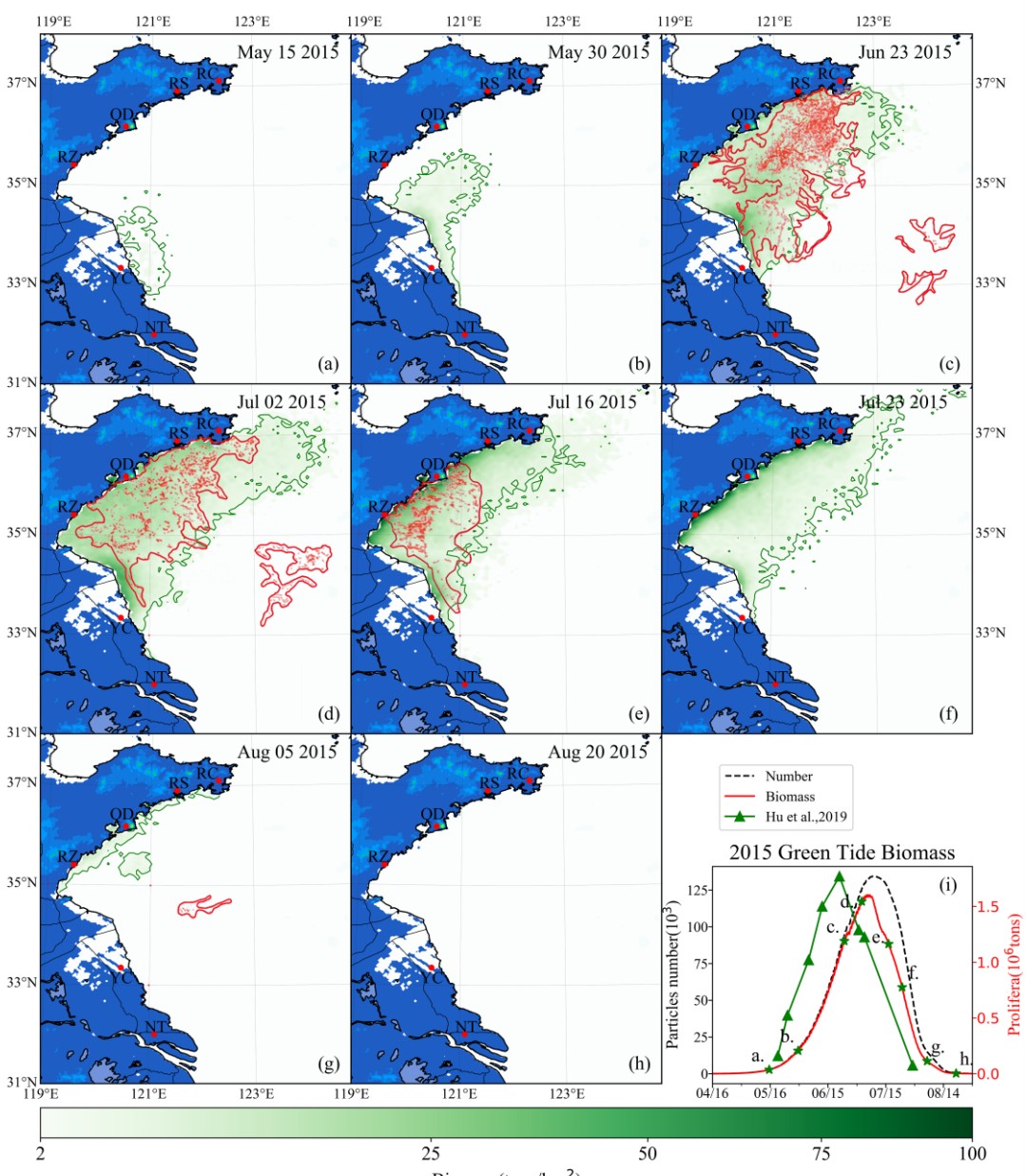

Figure 11. Comparison between simulation and remote sensing observation of green tides from May to August 2015 (a–h). The green image indicates the simulated biomass density of the green tide (color image, unit: tons/km$^2$). The red image shows the satellite-derived spatial coverage of *U. prolifera* from MODIS-TERRA. Panel (i) is the time series of simulated biomass and particle number of green tides in 2015, compared with observed biomass from Hu et al. (2019). The green pentagrams in panel (i) indicate the biomass of the corresponding date in the panel (a–h).

## 4 Discussion

### 4.1 Uncertainties of physical, biological, and anthropic processes

The observation of the entire bloom process is technically complex in the study of massive floating macroalgal blooms. In this study, a floating macroalgae growth and drift model was established, supplemented by remote sensing observations, which can reproduce the entire process and predict the development of macroalgal bloom. However, there are many uncertainties throughout the blooming process, which could significantly limit the precision of long-term prediction.

In the realistic numerical simulations of green tides from 2014 to 2015, the initial biomass of *U. prolifera* had the same
configuration of 4,800 tons, deployed continuously from April 16 to May 15 base on the estimation in previous studies (Liu et al., 2013; Xu et al., 2014a). The initial distribution was also uniform. However, high uncertainties regarding the biomass and distribution were observed. The initial biomass of *U. prolifera* was determined primarily by the scale of local *Porphyra* aquaculture around the Subei coastal region and the timing of harvest activities. The precise estimation of initial biomass and timing requires extensive monitoring for these activities, as well as robust and timely satellite assessment of satellite remote
sensing.

From the satellite observations in June and July 2014, we observed stable patches of *U. prolifera* off the Subei Shoal (Fig. 10c–g), indicating the continuous supply of *U. prolifera* from the local *Porphyra* aquaculture activities in summer, resulting in stable bloom off the Subei Shoal and northward drift. Therefore, this factor, which could lead to significant bias of *U. prolifera* distribution and biomass, should be considered during long-term simulation.

During the green tide bloom, large-scale salvage operations were implemented to reduce the biomass of floating *U. prolifera* in Jiangsu and Shandong coastal waters (Liu et al., 2013; Wang et al., 2018), which could significantly change the local biomass. The biomass of salvage operations reaches $1.5–2.0 \times 10^6$ tons every spring and summer along the Shandong coastal region (Ye et al., 2011; Zhou et al., 2015), which could be the reason for the underestimation of biomass from June 2–16, 2015 (Fig. 10d–e). The salvage operations cause significant uncertainty for numerical prediction, particularly along the coast where
the operations are primarily conducted.

The propagules are distributed near the floating *Ulva* with a high density and move with ocean flows (Li et al., 2017). The modified clay (MC) at a proper dose can flocculate with microscopic propagules and effectively remove microscopic propagules from the water column (Li et al., 2020). The physiological processes of *Ulva* cells could be disrupted by MC (Zhu et al., 2018). This method was frequently used to mitigate blooms in local areas (Li et al., 2017). The intervention of human
activities on the blooming process was not considered in the model. Large-scale salvage and elimination activities play essential roles in reducing the scale and intensity of the green tide bloom. When the observed biomass peaked, the biomass in the simulation maintained an increasing trend. Finally, the maximum simulated biomass was similar to the maximum estimated biomass.

### 4.2 Short-term variations and quick response

To reduce the errors of long-term simulation, caused by the complex origin of initial floating macroalgae and the uncertainty of growth and drift, the time of each simulation was shortened by dividing the entire long-term simulation into multiple short-term simulations and renewed the location and biomass in every short-term modeling by initialization of floating estimated by remote sensing observation.

Two consecutive simulations were carried out during the heyday of the YS green tide. One was configured for simulations
from June 15–23, 2014, and the other from June 23–30, 2014. According to the distribution from remote sensing observation and estimated biomass from Hu et al. (2019), the initial biomass and distribution of *U. prolifera* on June 15, 2014, was determined as shown in Fig. 12a and June 23, 2014, in Fig. 12d. The initial biomass was approximately 0.83 million on June 15 and approximately 0.96 million on June 23.

The time interval between two consecutive cloud-free satellite observations of green tides was generally large. Two intermediate results between the satellite observation gap were shown in Figs. 12b and 12e. After nearly one week of simulation, the coupled model system made precise simulation, compared with remote sensing (Fig. 12c, f), and the biomass was similar to that estimated via satellite remote sensing (Fig. 12g). Moreover, spatial distribution was well predicted. Compared with long-term simulation, the variation of green tide distribution and biomass could be determined more accurately by the results of the short-term simulation. The accuracy of short-term simulations is reliable, and the short-term prediction of floating macroalgal blooms can be achieved by combining the numerical model with the satellite observations.

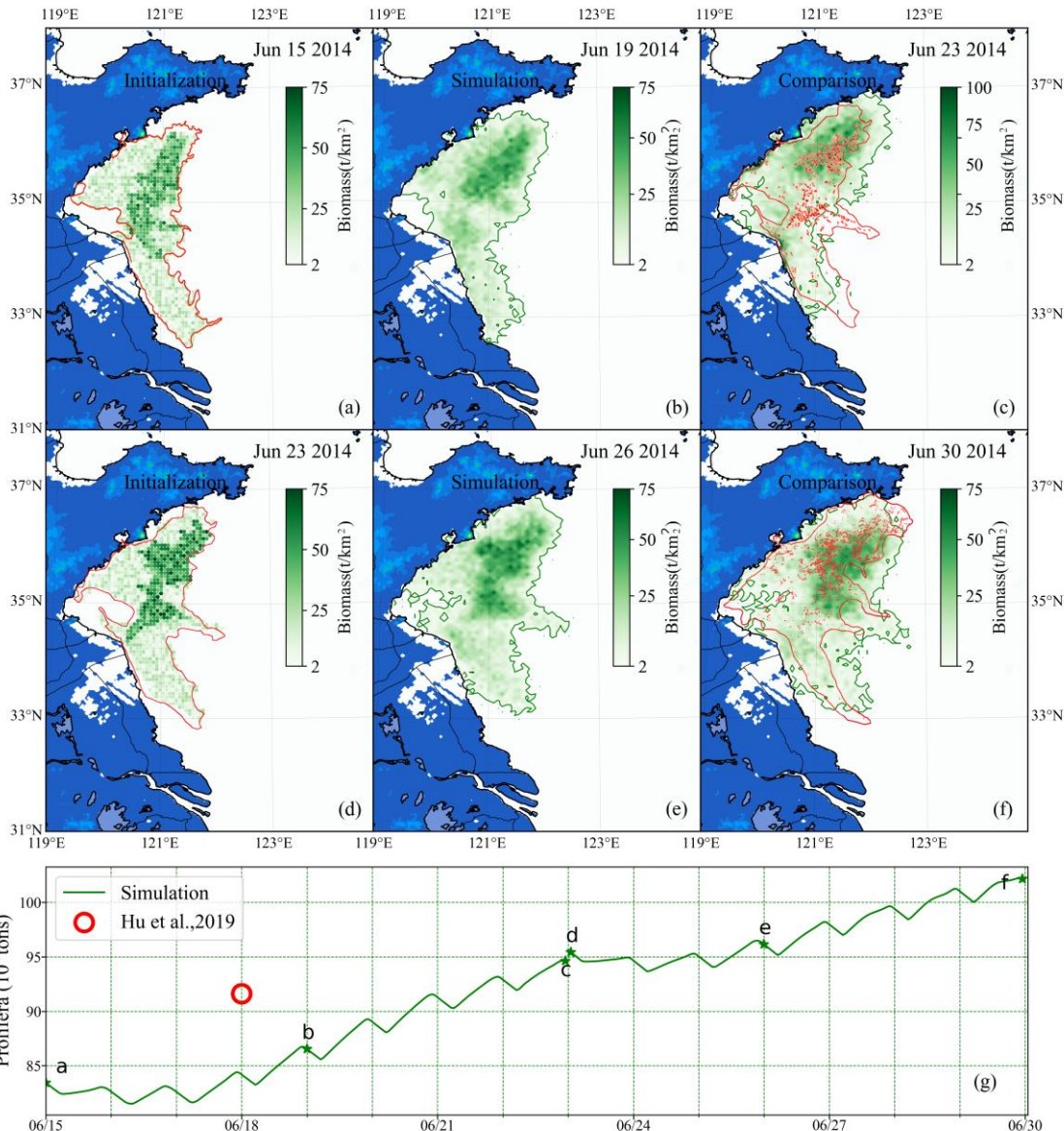

Figure 12. Comparison of short-term simulation and satellite observation of green tides in 2014: position and distribution of green tide density released initially: 15 June (a) and 23 June (d) based on the satellite data and estimated biomass. Panels (b) and (e) are the modeling results of 19 June and 26 June, respectively. Panels (c) and (f) compare simulation and satellite observation on June 23 and 30, respectively. The green image indicates the simulated result, and the red one indicates satellite observation. Panel (g) is the comparison of two consecutive simulated biomass (green line) and estimation biomass of satellite data (red circle). The green pentagrams indicate the biomass of the corresponding date in panels (a–f).

## 4.3 Roles of initial biomass, biotic and abiotic factors

The existence of diverse origins and continuous input of floating propagules significantly challenge the precise prediction and effective control of massive floating macroalgal blooms. In addition to the extensive provision from *Porphyra* aquaculture rafts in the Subei Shoal, the somatic cells, indicated by a laboratory study, could overwinter and restore growth on the annual spring bloom (Zhang et al., 2009), which is another significant source of *U. prolifera*. Additionally, four overwintering *Ulva* propagules that existed in sediments, including *U. prolifera*, may recover their growth when the temperature and irradiation are appropriate (Liu et al., 2012). Every April, before the occurrence of green tides, *Ulva* propagules are already widespread on the southern coast of the YS (Yuanzi et al., 2014). The transport trajectory was strongly affected by the origin of *U. prolifera*. Under the same environmental conditions, the scale of the bloom was determined primarily by the initial organisms. During the macroalgal bloom, the propagules supply from the coastal waters is continuously uncertain and difficult to determine through satellite observations or *in situ* surveys. Therefore, the feature that there was still large-scale *U. prolifera* distribution around the Subei Shoal in June and July 2014 has not yet been captured, as shown by satellite observations (Fig. 10), despite the continuous entering during the period of *Porphyra* aquaculture rafts collection (mid-April to mid-May) has been considered in the green tide simulations.

The growth of floating macroalgae is affected by a variety of environmental factors. The influences of abiotic factors (e.g. temperature and irradiation) and biotic factors (e.g. nutrients) have been considered in the macroalgae growth module. Many macroalgae species have a clear thermoperiod, which is sensitive to temperature. The changes of temperature in the surface layer of water column in the YS has significant seasonal characteristics (Figs. 5-6). From spring to late summer, the temperature controls biological processes of the main species of YS green tides from germination, growth, reproduction to extinction (Fan et al., 2015), which is reflected in the temporal dynamics of biomass (Figs. 10i and 11i). The photosynthesis is limited by light attenuation caused by self-shading and floating depth of macroalgae, which changes with the growth stage (Ren et al., 2014) and the module has not yet detailed this process. Meanwhile, the coastal turbid water that contains abundant suspended particulate matter can also limit the growth of green tide due to strong light attenuation in the upper surface water column. The initial release zone in Subei Shoal is influenced by significant suspended sediment dynamics (Bian et al., 2013). Therefore, the growth rate in the coastal region in Jiangsu and Shandong is probably limited by suspended sediment.

Nutrient eutrophication frequently results in macroalgal blooms in coastal waters (Liu et al., 2013), which can also be reflected in the simulation results. There are significant differences in the concentration of dissolved nutrients between the coastal and offshore areas of the YS (Fig. 7). The simulation showed that massive macroalgal bloom in the coastal of Subei (34°N) in late June (Figs. 10c, d, e, and 11c), which was directly related to the nutrient eutrophication here, with the dissolved nitrate about 8 mmol/m$^3$ and the dissolved phosphate over 0.6 mmol/m$^3$. The macroalgae growth in the offshore areas is relatively weaker because of nutrients limitation. The nutrient concentration is one of the major factors that influence spatial coverage of macroalgal bloom, and the abiotic factors including temperature and solar irradiation mainly modulate the temporal dynamics. Due to the difficulty in obtaining the distribution and variations of observational or simulated nutrients datasets, the accuracy of macroalgae simulation was limited by the deviation of nutrients datasets. Floating *U. prolifera* can efficiently absorb nutrients (Luo et al., 2012a), and the concentration of nutrients in the sea would decrease sharply when *U. prolifera* blooms dramatically, which may hinder the rapid growth of *U. prolifera* (Wang et al., 2019). When the green tide bloom reached its peak with millions of tons of biomass or drifted to the regions far from offshore, the dissolved nutrient concentration may be a significant growth limitation, even if the temperature and irradiation are still suitable for growth (Figs. 5 and 6). Due to lack of further research data on ecological relationships, some biotic factors (e.g. dissolved oxygen and potential biological competition against *Sargassum*) and abiotic factors (e.g. suspended particulate matter) are difficult to be parameterized (Solidoro et al., 1997), and therefore not considered in the physical-ecological coupled module.

**4.4 Prospects on model development**

No technique was identified for the precise quantification of the biomass of floating macroalgae (Sun et al., 2020). Most growth models only considered the environmental factors in a fixed station and disregarded the spatial variation of floating growth. The environmental factors vary significantly at different locations. Based on Lagrangian particle tracking, each particle was considered an independent simulation unit. The drift velocity and growth rate for that particle were obtained according to the natural environmental factors corresponding to the spatial position and time that particles locate. The simulation principle of

this model is suitable for the actual situation of massive floating macroalgal blooms, which float and grow across vast regions.

The large-scale bloom of floating macroalgae affected the distribution of nutrients. The simulations in future studies should incorporate the circulation of nutrients between macroalgae and the ocean environment to improve the coupled model development at a more precise spatiotemporal scale. By coupling with the regional ecosystem or biogeochemical model, this

model can be used to study the consumption of nutrients by the macroalgal blooms and its limitation on the growth of macroalgae. In particular, the model of floating *U. prolifera* could be established as a warning system of green tide disaster forecasting and be an efficient and economical tool for the prevention and management of green tides. Despite being used to simulate the green tide, this coupled model can also be applied to other large-scale macroalgae disasters that bloom in different parts of the world.

**5 Conclusion**

A system that coupled the ecological dynamic growth module with the physical drift module for macroalgae was developed to study the spatial and temporal variations of massive floating macroalgal bloom. The dynamic process of growth and drift is achieved by the replication/extinction and Lagrangian-based particle-tracking. It was applied to the dynamic simulation of the YS green tide blooms in 2014 and 2015, with environmental drivers from ECS-FVCOM. The simulation results were verified

against various observation data and demonstrated reasonable prediction precision. The modeling experiments also suggested that the surface wind played a crucial role in the northward drifting of *U. prolifera* from the Subei Shoal and finally resulted in an annual ecosystem disaster for the adjacent coastal region. The realistic simulation for two years exhibited many uncertainties from natural and human processes during the long duration from early spring to late summer, potentially leading to extensive prediction bias. However, the short-term simulation in this model, along with the determination of spatial coverage

and biomass, proved to be an efficient and robust system for accurate forecasting of the development of *U. prolifera*.

Although this unique tool for macroalgae prediction was only applied in the simulation of the YS green tide, it can potentially be used to study other macroalgal blooms, such as golden tides caused by *Sargassum*, in other regions where sufficient information on the macroalgae physiological relationship with environmental factors are available.

**Code and data availability**

The Fortran code of FMGDM v1.0 is available at https://doi.org/10.5281/zenodo.4459922 (last access: 3 July 2021). The example of the green tide in the Yellow Sea, China, is available at https://doi.org/10.5281/zenodo.4607828 (last access: 3 July 2021). The ECS-FVCOM forcing data (surface wind, radiations), the ocean bathymetry, and the results of ECS-FVCOM (water velocity, temperature, and nutrients), which are also used as input variables of FMGDM for the green tide simulation, the information initial particles position, the satellite pictures of the green tide in the YS, 2014 and 2015, and the drifter

trajectories dataset used to evaluate the tracking module are available at: https://doi.org/10.5281/zenodo.4616462 (last access: 7 July 2021).

## Author contribution

JG proposed and led this model development study. JG and FZ developed the coupled model. DL provided many important suggestions for this study and key data of *U. prolifera* growth. JG, PD, and CC contributed to the simulation result analysis of the ECS-FVCOM, which is used for this research. XW contributed to the remote sensing interpretation. FZ processed the model outputs and wrote the manuscript with contributions from all co-authors.

## Acknowledgments

This research is supported by the National Key R&D Program of China (Grant No. 2016YFA0600903) and the National Natural Science Foundation of China (Grant No. 41776104; 41761144062).

## Competing Interests

The authors declare that they have no conflict of interest.

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
