# Peer review of "A Lagrangian-based Floating Macroalgal Growth and Drift Model (FMGDM v1.0): application to the Yellow Sea green tide"

_Geoscientific Model Development, 2021_

## Author Response (AR1)

**Author response to the referees' comments**

Dear referees and editor,

We are very grateful for your positive comments and constructive suggestions on our manuscript "*A Lagrangian-based Floating Macroalgal Growth and Drift Model (FMGDM v1.0): application in the green tides of the Yellow Sea*". Following these comments and suggestions, we have revised this manuscript. In particular, We 1) added additional validations for the Lagrangian tracking module through a comparison with the observed surface and subsurface drifters, 2) upgraded the macroalgal growth model with the inclusion of nutrient's influences, 3) made the adjustments of initial deployments of *U. prolifera* around its origin region, and 4) re-ran all cases and updated all results based on these re-configurations.

Our response to the comments of each referee and corresponding changes are provided in this document, with a **bold** front for comments and a regular front for our answers. The manuscript quotations are represented in *italics*, and the changes in the manuscript are highlighted as underlined. The introductory parts of the referee report are omitted and marked as […] to keep the document concise.

We hope that our response can address the referee's concerns sufficiently.

Sincerely,

Fucang Zhou and Jianzhong Ge (on behalf of the author team)

**Referee comment #1**

**[…] Although some bias still existed, this is acceptable considering the complex influencing factors in the system. What is more remarkable is that this model can simulate the coverage and biomass of the floating macroalgae much better when using short-term modeling. This is pretty useful for predicting when green tides are happening.**

Thanks. We have significantly improved the manuscript to address the reviewers' comments. For details, please see the responses to the other two reviewers below.

**Referee comment #2 and #3**

From comment #2, part 1:

**[…] Yet it fails to capture the year to year variability seen in the satellite product – in magnitude and extent –, suggesting important drivers of U. prolifera are not represented. Overall, the structure of the paper could be improved, so does the writing as abbreviations and typos are presents. I do not think that the experimental set-up is sufficiently convincing to ascertain that these results are robust.**

1. **the validation of the Lagrangian transport for supporting the choice of ocean model and fraction of the wind component for the windage is not robust; the authors set windage on 3.5% without having completed a sensitivity analysis or referencing a study that did so. Macroalgae are influenced by wind yet the fraction depends on the physical characteristics of the floating object. The passive drifting experiments should be carried out with a much larger number of particles in the order of tens of thousands (currently only 6), and released over a wider period (currently simply a single date instead of across a full month of prolifera observed presence in the release area). Density maps of particle positions after 120 days for each experiments (ocean currents only, and different windage factors) should be provided to illustrate the differences in drifting patterns, together with a thorough**

**argumentation for the choice in the balance of ocean current and windage for U. prolifera drifting. I also suggest the use of the Global Drifting Program dataset (if sufficient floats are available in the area) to evaluate drifting patterns in the region using drogued vs undrogued surface floats (that remove or include a wind factor), and evaluate the model skills in reproducing the patterns from observations. Such work would also benefit the interpretation of the patterns obtained with the individual based growth model, in order to disentangle influence of biology and physics to explain biomass distribution.**

**From comment #3, part 1:**

**[…] However, the paper is missing some important details and background information. Here are some specifics:**

1. **Windage is assumed to be 1.5% to 3.5%, but the authors do not explain why this value was taken. As the authors pointed out with the simulations without wind, wind plays a crucial role in determining trajectories. Please cite references and detail your reasoning behind this selection. The authors should determine if the surface layer of the hydrodynamic solution already represents some movement due to wind. The authors may consider looking at analogous studies with large floating objects (e.g., tsunami debris) or drifter studies to calibrate windage.**

Our responses to the uncaptured annual variability of green tide and our attempt to solve this problem are given in our replies to the referrer's comment #2, part 4. Here, we first respond to the comment about windage, which both referees raised. Then, following the referee's suggestions, we have conducted a series of experiments on windage range in 2.7-3.5% and added the experimental results in Section 3.2.2. Additionally, more supporting references were cited in the revision to select a more appropriate windage coefficient.

To reduce the computational load, we used the super-particle approach. A super-particle was defined as an ensemble particle accounting for 100 tons of *U. prolifera*. This approach is commonly used in larval transport studies. Examples can be seen in recent scallop larval transport studies done by (Chen et al., 2021). Considering the rapid algal growth during the bloom outbreak, the individual number of *U. prolifera* could increase up to >15000 at the peak of the bloom. In the revision, we have increased the initial number of particles to 480, 10 times more than the number used in our last submission. To better compare drifting patterns under different windage, we remain the initial particle release on May 1. In the realistic simulations of green tides, we released the particles continuously over an entire month, as the referees suggested.

Regarding the validation of particle tracking in the Yellow Sea and adjacent regions, we checked the Global Drifting Program dataset to search the data in the region. Unfortunately, the dataset does not cover our study areas. Luckily, we found some drifter trajectory data in our region from Bao et al. (2015). We used these data to evaluate the skills of our particle tracking algorithms and the accuracy of the hydrodynamic model ECS-FVCOM. In addition, we also conducted a drogued-drifter experiment in the nearby East China Sea area over the summer of 2017. The results of these additional studies were added in the revision to strengthen the model validation. See the detail below.

By the way, the hydrodynamic surface layer had already included a 1.5% wind effect, and the other wind drags for particles drifting were composed by direct windage.

The modification contained two parts in the revision: a) the windage experiments and b) the tracking model validation.

   a)   The revised windage experiments.

In section 2.7, the model configuration is revised.

[revised manuscript text omitted]

The revision of tracking module evaluation:

**2.5.3 Drifting trajectory data**

The drifting dataset used to evaluate the skills of the tracking module is composed of two parts: the trajectory data of satellite-tracked surface drifters released from Subei Shoal in 2012 (Bao et al., 2015), and the subsurface drogued-float tracking data in the inner shelf of the East China Sea in 2017. All the original trajectory data are available at: https://doi.org/10.5281/zenodo.4616462.

**3.2 Validation of Tracking module**

**3.2.1 Tracking module evolution**

A total of seven satellite-tracked drogued-drifters were used to evaluate the skills of the particle tracking model, including five surface drifters released in the YS (Bao et al., 2015) and two subsurface drifters released in the ECS. The surface drifters contained four 40-cm width, 70-cm height rectangular sails, and a large central buoy (Bao et al., 2015). The subsurface drifter was constructed by a 67-cm diameter, 6-m height cylindrical subsurface sail, and a 28-cm diameter central buoy.

Seven particle-tracking simulations were conducted. One hundred particles were released at a location that matched the drifter's in-situ deployment position, and the horizontal random diffusion coefficient ($K_r$) was set as 50 $m^2/s$. In addition, the depth for surface and subsurface drifters were set as 0.5 m and 2 m, respectively. Thus, for these drogued-drifters, only half of the buoy was exposed above the sea surface, and the direct wind factor was not considered in the tacking simulations.

The simulated particle trajectories are generally consistent with the observed drifter trajectories, particularly for a short-term prediction (Fig. 8). The tracking time for surface drifters #1–5 lasted more than one month (Fig. 8a–e). The results show that the model was robust to reproduce the overall drifter's movement directions. Since the drifter paths may change with strong randomness due to complex variations of ocean flow, winds, and waves, the long-term prediction of drifter paths and dispersal could be of great challenge. The tracking time for subsurface drifters #6–7 (Fig. 8f) was relatively shorter, only 5–9 days. Compared with surface drifters, subsurface drifters were driven by a more complicated forcing relating to the large depth range of the sail. The water flow at a 2-m depth was selected as the driving force approximately, considering the average drifting state of subsurface drifters. The simulated trajectories were similar to the observed drifter's movement trends. However, the drifted distance was slightly shorter than the actual situation.

The model-data compression suggests that the particle tracking algorithm in FMGDM can provide reasonable predictions for free-floating drifters with higher confidence for surface drifters. In addition, the hydrodynamic model, ECS-FVCOM, is reliable. Our results showed that U. prolifera were mainly under a free-floating state at the sea surface.

[Figure]

*Figure. 8. Comparisons between observed (red lines) and simulated (grey lines) drifter trajectories.*

**Referee comment #2 and #3**

From comment #2, part 2:

2. **The prolifera drifting experiment is carried out with an insufficient number of particles to capture the full ocean model variability, failing to convince the reader that the patterns obtained are robust. What's the rationale for deploying only 47 particles (each representing 100 tons of biomass), i.e. at 0.1° resolution on a single date (1st May), instead of for example deploying a particle in every gridcell of the model (resolution of 0.5 to 3km), together with a lower biomass per seeded particle, in that region, and repeated across the full period when U. prolifera is present in the area (mid-april to mid-May)? Are the results numerically stable with such low number?**

From comment #3, part 3:

3. **Model initialization is also important. The authors note a May 1 start time, several starting locations, and that 47 particles were used. More detail needs to be provided as to why the model was initialized this way. For example, 47 is far fewer particles than are typically used with trajectory modeling, so was computational cost the reason for so few particles?**

**Response**: Regarding the initial particle number (47), see our explanation above. We re-did the 2014 green tide simulation with ten times more particles at initial (480, each representing 10 tons of biomass). These particles were deployed evenly in the same release area (resolution of ~3 km) with settings as the same as those in the previous submission. It should be noted that the particle number increased during the growth process, reaching more than 1.8 million tons at the bloom peak (Fig. R1). It was nearly 400 times more than that at the initial state. Comparing with Fig. 9 in the last submission, there are no significant differences in the total biomass and the spatial patterns between the two simulation results.

[Figure]

Figure R1 (for response). The 2014 green tide simulation, with 480 initial particles deployed with a separate scale of ~3 km.

We agree with the referees that adding simulated particles number is helpful to resolve the spatiotemporal flow variability. In subsequent simulations, the initially deployed particles were added by ten times, and each particle contained 10 tons of biomass. The selection of particles number was determined to have a balance between simulation accuracy and computational efficiency. More details could be found in the revision below. The settings of *U.prolifera* simulation were revised in section *2.7 model settings*

*Most importantly, two realistic dynamic growth simulations were conducted. To verify the general applicability of the model, we simulated the growth and drift processes of U. prolifera in the YS in 2014 and 2015, respectively, with identical model configurations. In the two simulations, each particle represented 10 tone biomass of floating U. prolifera, so that 4,800 tons were deployed initially. The initial coverage and biomass of the U. prolifera were determined based on the field surveys by Liu et al. (2013) and Xu et al. (2014a). The simulation time was 135 days from April 16 to August 28. The initial particles deployed continuously from April 16 to May 15, 2014. Daily 160-ton biomass was spatial randomly released in the hot-spot zone (Fig. 3b) over an entire month. In this study, instantaneous environmental factors, including temperature, nutrients, solar radiation intensity, ocean flow, and wind speed, were determined from the physical ECS-FVCOM model.*

**Referee comment #2**

3. **The ecological model could be better explained, in relation to the external factors chosen (and those disregarded) and enable the reader to get an understanding of how these factor influence growth/death. An equation of the prolifera dynamics over time as a function of these factors is needed. It seems though, that the model dynamics is simply driven by the laboratory-based measured rates for different ambient conditions, and that no physiological model exists. If this is not the case, it should be better explained, together with a sensitivity analsysis of the model parameters. Nutrients are known to influence macro algae growth, why is nutrient uptake not considered, using for example a biogeochemical model like in Brooks et al. (2018, Marine Ecology Progress). Only temperature causes mortality. What's the need for including salinity?**

We greatly appreciate this excellent suggestion. We have upgraded our ecological module in the revision by considering nutrient uptake in macroalgae growth. The literature provided by the referee, Brooks et al. (2018), developed an ecological model for *Sargassum*. Unfortunately, it is not applicable for the growth of *U.prolifera*. We found two studies in the ecological dynamic growth simulation of *Ulva* sp., which are more suitable for the ecological simulation of green tides (Ren et al., 2014; Sun et al., 2020). We adopted methodology from these studies to upgrade the ecological module in this revision. The physiological process was controlled by the function of temperature, irradiation, and nutrients. The original purpose of considering the salinity effect was to limit the growth of *U.prolifera* in low-salinity water., i.e., in the Changjiang River where the salinity was less than 25 PSU. However, this situation rarely occurred. The *U.prolifera* mainly grew in the high-salinity region. For this reason, the salinity limitation was removed from the biological module.

The revision of the section *2.1 Model framework*:

[revised manuscript text omitted]

$$f_m(T) = \begin{cases} 0, T \le 25.7°C \\ 0.01416T^3 - 1.223T^2 + 35.22T - 337.73, T \ge 25.7°C \end{cases} \qquad (20)$$

According to the floating growth characteristics of U. prolifera, the self-shading limited function $f(\rho)$ was determined. When the assembled density does not exceed 0.16 mol C/m², the growth of U. prolifera is not restricted by self-shading. However, as the density increases, the accumulation of U. prolifera becomes significant, and maximum when the density is greater than 0.56 mol C/m².

$$f(\rho) = \begin{cases} 1, \rho \le 0.16 \\ 2.308 \ exp(-2.5\rho) - 0.54705, 0.16 < \rho \le 0.56 \\ 0, \rho > 0.56 \end{cases} \qquad (21)$$

[revised manuscript text omitted]

**Referee comment #2**

4.  **The model reproduces the bloom dynamics to first order, yet little year to year variability is present in the spatial patterns for the two years of the study, although a clear inter-annual signal exists in the observations. For instance large parts of prolifera patches in July 2018 are uncaptured by the model. It is not possible to determine what is the cause for these dicrepancies (biological or physical?). I wonder whether a better choice of particle deployment at start (as mentioned above) would improve the results. In support, we see that short term simulation (1 week) show better skills. Could this be explained by a better initialisation? An analysis (or inverse modelling approach)**

**would be useful to get a feeling of the sensitivity to the choice of parameter values (whether physical or ecological) for best recreating the observed biomass concentrations.**

In this study, we only simulated the bloom of YS green tides from spring to summer, assuming all green algae die in late summer. However, according to the related studies (Liu et al., 2012; Zhang et al., 2009), limited green algae could overwinter and restore growth in next spring in offshore and central YS. Therefore, the correlation between two consecutive years was limited but still exists. Meanwhile, the annual variability in the bloom biomass and spatial patterns significantly correlates with the initial biomass and origin place in different years. Limited by the difficulty of comprehensive in-situ investigation and consequently lack information for biomass and coverage, we had to make the simulation for 2014 and 2015 with the same initialization of biomass and spatial patterns. We determined the initial biomass of about 5,000 tons by Liu et al. (2013) and Xu et al. (2014a) and deployed in the coast of Yancheng and Nantong, the main origin areas of *U. prolifera*.

About the large parts of *U.prolifera* patches uncaptured by the model, it should be the results in July 2014. We had discussed that in the section as follows:

*4.3 Roles of initial biomass and nutrient limitation*

*The existence of diverse origins and continuous input of floating propagules significantly challenge the precise prediction and effective control of massive floating macroalgal blooms. In addition to the extensive provision from Porphyra aquaculture rafts in the Subei Shoal, the somatic cells, indicated by a laboratory study, could overwinter and restore growth on the annual spring bloom (Zhang et al., 2009), which is another significant source of U. prolifera. Additionally, four overwintering Ulva propagules that existed in sediments, including U. prolifera, may recover their growth when the temperature and irradiation are appropriate (Liu et al., 2012). Every April, before the occurrence of green tides, Ulva propagules are already widespread on the southern coast of the YS (Yuanzi et al., 2014). The transport trajectory was strongly affected by the origin of U. prolifera. Under the same environmental conditions, the scale of the bloom was determined primarily by the initial organisms. During the macroalgal bloom, the propagules supply from the coastal waters is continuously uncertain and difficult to determine through satellite observations or in situ surveys. Therefore, the feature that there was still large-scale U. prolifera distribution around the Subei Shoal in June and July 2014 has not yet been captured, as shown by satellite observations (Fig. 10), despite the continuous entering during the period of Porphyra aquaculture rafts collection (mid-April to mid-May) has been considered in the green tide simulations.*

As the referee mentioned, a better choice of particle deployment as initialization would improve the results. Remote sensing observation was the most effective way to get the spatial patterns of the green tide. The short-term simulation showed better skills when the model was initialized with the spatial distribution of green tide by remote sensing data.

The inverse tracking could provide a reasonable estimation of position for a short time. However, with 1–2 month inverse tracking, the model still fails to estimate the origin place of the green tide, covering a significantly large area. This overestimation of origin place is more significant when considering the random walk process in the tracking algorithm. We have tried to deployed initial particles continuously across an entire month (mid-April to mid-May). However, the *U.prolifera* floating in the Jiangsu region in July was still difficult to reproduce, which suggests there is still a possible *U. prolifera* source near Subei Shoal even in the summertime of July and August.

**Referee comment #3**

2. **Most trajectory models also include dispersion to represent forcings not caused by wind or currents. This dispersion is often a stochastic component. If the authors determine not to include dispersion in their model simulations, there should at least be acknowlegement of the absence of dispersion and resulting implications.**

The random dispersion had already been included in the model, which was considered as horizontal and vertical random walk by adding extra terms to particle trajectory calculation. Since the *U. prolifera* mainly floats at the sea surface without the vertical migration, the vertical random walk was disabled in the model setting. We have clarified that in the revision to better highlight the model capability and configuration on random dispersion as follows:

[revised manuscript text omitted]

---

## Referee Report (RR1)

The authors with their revised manuscript have addressed all points raised during the initial process. The structure is much improved and makes the manuscript easier to read. The inclusion of a validation for the drifting model, detailed aspects for the choice of windage, better particle initialisation across a larger time window, and a nutrient component in the growth model, make the results more robust.

All these changes (% windage, nutrient component, repeated particle-release across time) lead to a better representation of the bloom, with the attenuation/killing of the eastern propagation of the bloom from 28th July onwards for 2014, despite still not capturing the southern tail present in the observations mid-end July. Much like 2014, a better attenuation/killing of the bloom is obtained early august.

The changes also affect the level of biomass for the short-term monitoring experiments, with the revised growth model showing a much better fit with observations by being less dynamic (+ $10.10^4$ tons in two weeks for the 15 and 23rd of June 2014 experiments, respectively – compared to +$50.10^4$ tons, and +$35.10^4$ tons in the initial setup). These results show a clear influence of the added components (nutrients?), yet I feel this could be discussed a bit more; an extension of this study may include some parameter fitting (I understand this may not be in the scope of this study), as It remains difficult to disentangle the impact of biotic (nutrients) and abiotic (temperature) factors on the bloom spatial and temporal dynamics e.g. propagation/attenuation of the bloom.

The quality of the writing has vastly improved, but some typos remain.

---

## Author Response (AR2)

**Author response**

Dear editor and referees,

Thank you for your recognition of our revised manuscript "*A Lagrangian-based Floating Macroalgal Growth and Drift Model (FMGDM v1.0): application to the Yellow Sea green tide*".

Following the suggestions of Referee #2, we 1) added additional discussion of the impact of biotic and abiotic factors on macroalgal bloom to Section *4.3*, 2) revised the typos, grammatical errors, and figure labels in our manuscript.

Our response and corresponding changes are provided in this document, with a **bold** front for comments and a regular front for our answers. The manuscript quotations are represented in *italics*, and the changes in the manuscript are highlighted as underlined. The introductory parts of the referee report are omitted and marked as […] to keep the document concise.

We greatly appreciate all comments and suggestions on this manuscript from referees. We hope that our response can address the referee's concerns sufficiently, and this revision can meet the standard of *Geoscientific Model Development*.

Sincerely,

Fucang Zhou and Jianzhong Ge (on behalf of the author team)

**Referee comment #2**

**[…] The changes also affect the level of biomass for the short-term monitoring experiments, with the revised growth model showing a much better fit with observations by being less dynamic (+10.10$^4$ tons in two weeks for the 15 and 23rd of June 2014 experiments, respectively–compared to +50.10$^4$ tons, and +35.10$^4$ tons in the initial setup). These results show a clear influence of the added components (nutrients?), yet I feel this could be discussed a bit more; an extension of this study may include some parameter fitting (I understand this may not be in the scope of this study), as It remains difficult to disentangle the impact of biotic (nutrients) and abiotic (temperature) factors on the bloom spatial and temporal dynamics e.g. propagation/attenuation of the bloom. […]**

The changes of the increased level of biomass in the short-term experiments, mentioned by Referees #2, were largely caused by the nutrient limitation for macroalgae photosynthesis in offshore areas.

As the Referee #2 commented, it is difficult to disentangle the impact of biotic (nutrients) and abiotic (temperature) factors on the bloom spatial and temporal dynamics. The effects of biotic and abiotic on ecological dynamics of macroalgae, more specifically green tide of *U. prolifera,* have been extensively examined by previous modeling and laboratory studies (Cui et al., 2015; Wang et al., 2019). In this study, the major motivation of our proposed model is to determine the joint effect of biotic and abiotic on macroalgae growth.

Many biotic and abiotic environmental factors were considered in the growth simulation of the ecological module. The suggestions of the referee provide a good idea to understand the ecological dynamics of macroalgae. According to simulation results presented in this work, the biotic factors (nutrients) is one of the major factors that influence spatial coverage of macroalgal bloom, and the abiotic factors including temperature and solar irradiation mainly modulate the temporal dynamics. Following Referees #2's comments, we have added additional discussion of the impact of biotic and abiotic factors on macroalgal bloom to Section *4.3*.

1) The revised of the additional discussion:

*4.3 Roles of initial biomass, biotic and abiotic factors and nutrient limitation*

*[...]*

*The growth of floating macroalgae is affected by a variety of environmental factors. The influences of abiotic factors (e.g. temperature and irradiation) and biotic factors (e.g. nutrients) have been considered in the macroalgae growth module. Many macroalgae species have a clear thermoperiod, which is sensitive to temperature. The changes of temperature in the surface layer of water column in the YS has significant seasonal characteristics (Figs. 5-6). From spring to late summer, the temperature controls biological processes of the main species of YS green tides from germination, growth, reproduction to extinction (Fan et al., 2015), which is reflected in the temporal dynamics of biomass (Figs. 10i and 11i). The photosynthesis is limited by light attenuation caused by self-shading and floating depth of macroalgae, which changes with the growth stage (Ren et al., 2014) and the module has not yet detailed this process. Meanwhile, the coastal turbid water that contains abundant suspended particulate matter can also limit the growth of green tide due to strong light attenuation in the upper surface water column. The initial release zone in Subei Shoal is influenced by significant suspended sediment dynamics (Bian et al., 2013). Therefore, the growth rate in the coastal region in Jiangsu and Shandong is probably limited by suspended sediment.*

*Nutrient eutrophication frequently results in macroalgal blooms in coastal waters (Liu et al., 2013), which can also be reflected in the simulation results. There are significant differences in the concentration of dissolved nutrients between the coastal and offshore areas of the YS (Fig. 7). The simulation showed that massive macroalgal bloom in the coastal of Subei (34°N) in late June (Figs. 10c, d, e, and 11c), which was directly related to the nutrient eutrophication here, with the dissolved nitrate about 8 mmol/m³ and the dissolved phosphate over 0.6 mmol/m³. The macroalgae growth in the offshore areas is relatively weaker because of nutrients limitation. The nutrient concentration is one of the major factors that influence spatial coverage of macroalgal bloom, and the abiotic factors including temperature and solar irradiation mainly modulate the temporal dynamics.*

*The influences of nutrients in macroalgae growth have been considered in the model. Due to the difficulty in obtaining the distribution and variations of observational or simulated nutrients datasets, the accuracy of macroalgae simulation was limited by the deviation of nutrients datasets. Floating U. prolifera can efficiently absorb nutrients (Luo et al., 2012a), and the concentration of nutrients in the sea would decrease sharply when U. prolifera blooms dramatically, which may hinder the rapid growth of U. prolifera (Wang et al., 2019). When the green tide bloom reached its peak with millions of tons of biomass or drifted to the regions far from offshore, the dissolved nutrient concentration may be a significant growth limitation, even if the temperature and irradiation are still suitable for growth (Figs. 5 and 6). Due to lack of further research data on ecological relationships, some biotic factors (e.g. dissolved oxygen and potential biological competition against Sargassum) and abiotic factors (e.g. suspended particulate matter) are difficult to be parameterized (Solidoro et al., 1996), and therefore not considered in the physical-ecological coupled module.*

**References**

Cui, J., Zhang, J., Huo, Y., Zhou, L., Wu, Q., Chen, L., Yu, K., and He, P.: Adaptability of free-floating green tide algae in the Yellow Sea to variable temperature and light intensity, Marine Pollution Bulletin, 101, 660-666, https://doi.org/10.1016/j.marpolbul.2015.10.033, 2015.

Wang, C., Su, R., Guo, L., Yang, B., Zhang, Y., Zhang, L., Xu, H., Shi, W., and Wei, L.: Nutrient absorption by *Ulva prolifera* and the growth mechanism leading to green-tides, Estuarine, Coastal and Shelf Science, 227, https://doi.org/10.1016/j.ecss.2019.106329, 2019.